# Latent Reasoning in LLMs as a Vocabulary-Space Superposition

## Abstract

Large language models (LLMs) demonstrate strong reasoning abilities with chain-of-thought prompting, but explicit reasoning introduces substantial computational overhead. Recent work on latent reasoning reduces this cost by reasoning in latent space without explicit supervision, but performance drops significantly. Our preliminary experiments suggest that this degradation stems from the unstructured latent space, which makes fitting latent tokens difficult. To address this, we restrict the latent space to the column space of the LLM vocabulary, treating latent reasoning as a superposition over vocabulary probabilities. Once latent reasoning concludes, it collapses into an eigenstate of explicit reasoning to yield the final answer. Based on this idea, we propose Latent-SFT, a two-stage learning framework. In the first stage, we design two specialized attention masks to guide the Latent Token Encoder in generating latent tokens, allowing the LLM to produce the correct answer conditioned on them. In the second stage, the Latent Token Encoder is discarded, and the LLM is directly trained to generate these latent tokens autonomously for latent reasoning, optimized with KL and CE losses. Latent-SFT sets a new state of the art on GSM8k, matching explicit SFT performance while cutting reasoning chains by up to 4× and outperforming prior latent methods. On Math500 and AIME24, lexical probability–based latent reasoning also clearly surpasses hidden-state–based approaches. Our metrics of effective compression rate and effective global parallelism further show that latent reasoning is both the compression of a single path and the superposition of multiple paths.

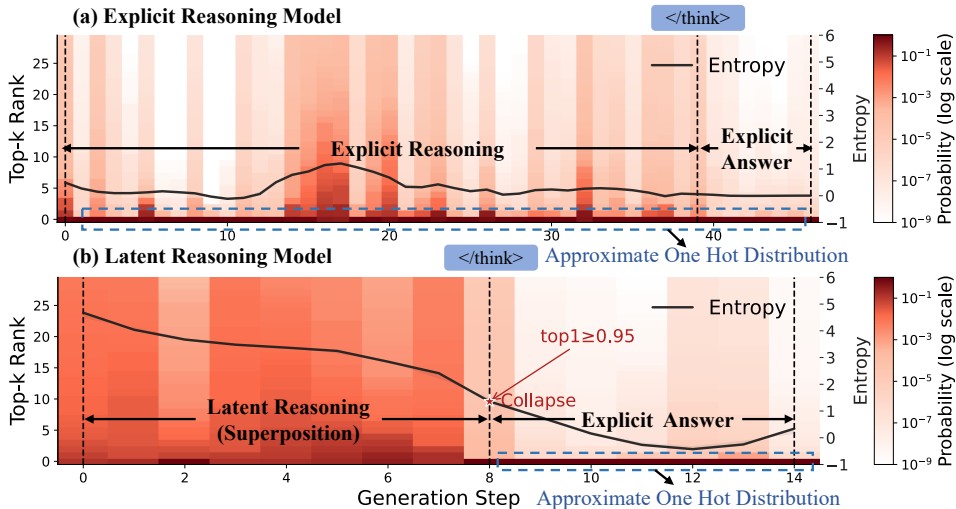

**Figure 1:** Comparison of Latent and Explicit Reasoning. Latent reasoning operates in a vocabulary superposition state, completing the reasoning process and reaching the correct answer with fewer tokens. Each step of explicit reasoning can be regarded as approximating a one-hot distribution.

## 1 Introduction

Large language models (LLMs) have shown strong reasoning abilities across diverse tasks. A key driver of this success is chain-of-thought prompting, which solves complex problems by generating

intermediate steps in natural language. However, current reasoning models rely on long chains of tokens, leading to redundancy and high computational cost. Existing solutions, such as token pruning (Xia et al., 2025) and reinforcement learning with length penalties (Arora & Zanette, 2025), reduce token usage but still constrain reasoning to the natural language space. In this work, we explore Latent Reasoning, a framework that enables reasoning in continuous latent space.

Unlike explicit reasoning, latent reasoning retains the last hidden state from the latent layer of the LLM, bypassing token sampling and directly feeding it back as the next-step representation. This hidden state is expected to carry richer information than a single token, effectively increasing bandwidth and enabling exploration of more reasoning paths with fewer generation steps (Zhu et al., 2025). While recent latent reasoning methods have succeeded in reducing token usage, they often suffer from substantial performance degradation (Hao et al., 2024).

To enhance latent reasoning, we analyze its challenges by comparing it with explicit reasoning. Firstly, supervision is difficult. Explicit reasoning leverages natural language as an anchor, enabling supervision through annotation or sampling data, with labels transferable across models. Latent reasoning, however, lacks natural labels; even if latent tokens are generated, they are often model-specific and hard to reuse. Secondly, learning is difficult. Explicit labels are discrete distributions, easy to optimize, and aligned with pre-training objectives. By contrast, latent tokens are typically the hidden states of the LLM's last layer—high-dimensional, unconstrained, and usually optimized via MSE or cosine distance, which diverges from the pre-training task. In summary, effective latent reasoning requires a principled definition of latent tokens that supports model learning.

In Section 3, we present preliminary experiments leading to two key observations. Observation 1: the hidden state distribution of the LLM's last layer is entirely inconsistent with the token embedding distribution. Since LLMs are trained with token embeddings as inputs, directly feeding hidden states introduces out-of-distribution values, preventing the model from interpreting their semantics and degrading performance. Observation 2: the effective rank of token embeddings in current LLMs is not full, indicating that the semantic space is inherently low-dimensional. Without constraining latent tokens, the model risks losing semantic consistency and struggles to learn meaningful representations. Based on these findings, we define latent tokens as elements within the column space of token embeddings. This ensures that latent and explicit tokens share consistent distributions and reside in the same low-rank space, thereby addressing the two central challenges revealed by our experiments.

Building on our definition of latent tokens, we introduce Latent-SFT, a two-stage learning framework. Stage 1: Latent token generation. We construct an encoder–decoder architecture using the same LLM, where the encoder is forced to produce latent tokens that enable the decoder to generate the correct answer. Specifically, the Latent Token Induction Mask (LTIM) constrains special tokens in the encoder to attend only to a sub-segment of the explicit reasoning chain. Latent tokens are then derived by linearly combining the vocabulary probabilities associated with these tokens. The Latent Token Supervision Mask (LTSuM) further enforces that each latent token decodes the subsequent explicit reasoning segment and the final answer. Together, these mechanisms ensure that latent tokens remain semantically compact, compatible, and correct. Stage 2: Latent token learning. The decoder learns to generate latent tokens independently. Specifically, KL loss is used to learn latent tokens, while CE loss is applied to optimize the final explicit answer.

As illustrated in Figure 1, our model performs latent reasoning in the column space of the vocabulary, analogous to wavefunction superposition in Hilbert space. Once reasoning is complete, entropy collapses into an eigenstate, producing the final natural language answer. Experiments confirm that Latent-SFT surpasses other latent baselines and achieves performance comparable to explicit SFT models on low-difficulty mathematical reasoning datasets. On high-difficulty datasets, latent tokens defined via soft embeddings significantly outperform those defined by hidden states. Further analyses reveal that its latent reasoning process is not merely a compression of a single reasoning chain but also a superposition of multiple chains.

## 2  RELATED WORK

Our work centers on latent inference, and we categorize related research into two groups: training-based methods and training-free methods.

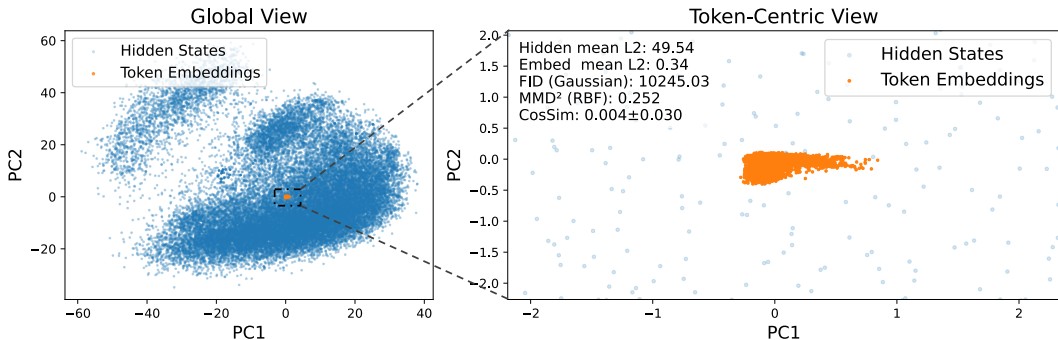

**Figure 2:** Visualization of last-layer hidden states and token embeddings of the LLaMA-3.2-1B-Instruct model on the GSM8k dataset. Results are shown from both global and token-specific perspectives to highlight the substantial distributional differences. In addition, we report the statistical differences between the two distributions using the Fréchet Inception Distance (FID) and the squared Maximum Mean Discrepancy (MMD$^2$). We also estimate cosine similarity via random sampling.

**Training-based Latent Reasoning.** COCONUT (Hao et al., 2024) first proposed using the last hidden state as the input for the next token without applying the language model head, laying the foundation for latent reasoning. However, it relied on a fixed number of latent steps and did not address the misalignment between hidden (latent) states and token embeddings, leading to weak performance. CCOT (Cheng & Durme, 2024) introduced continuous contemplation tokens and extended this idea to variable-length latent reasoning, achieving modest gains. CODI (Shen et al., 2025) improved alignment by adding an MLP to map hidden states into the token embedding space, and further adopted a teacher–student distillation strategy, forcing the latent model to match hidden states across all layers up to the final token of the explicit model—significantly boosting performance. PCCOT (Wu et al., 2025b) enhanced CODI by applying Jacobian iteration to improve both training and inference efficiency. CoLaR (Tan et al., 2025) advanced this line by introducing a latent head to predict compressed embedding distributions, maintaining alignment between latent and token spaces. Combined with reinforcement learning, CoLaR achieved state-of-the-art latent reasoning, though still trailing explicit SFT. Unlike these approaches, we restrict latent reasoning to the column space of the LLM vocabulary, directly addressing misalignment between latent and token spaces.

**Training-free Latent Reasoning.** Several works have explored training-free latent reasoning, including Soft Thinking (Zhang et al., 2025) and Mixture-of-Input (MoI) (Zhuang et al., 2025). Like our approach, they use next-token probability distributions to linearly combine token embeddings, thereby linking hidden and embedding spaces. However, Wu et al. (2025a) showed that this remains a form of greedy decoding, which limits performance. In contrast, our method can be seen as a trained variant of Soft Thinking, enabling the model to explore multiple reasoning paths.

## 3 PRELIMINARY EXPERIMENTS: DEFINITION OF LATENT TOKEN

To gain a deeper understanding ofing of the latent reasoning process, we conducted two preliminary experiments addressing the core question of what the representation of latent tokens should be.

### 3.1 DISTRIBUTIONAL DISCREPANCY BETWEEN FINAL-LAYER HIDDEN STATES AND TOKEN EMBEDDINGS

Most existing studies (Shen et al., 2025; Wu et al., 2025b) follow the COCONUT design, using the last-layer hidden state directly as the representation of the next latent token rather than sampling from the explicit token embedding matrix. However, a statistical analysis of the last-layer hidden states and token embeddings in the representative LLMs reveals the following observation.

**Observation 1.** *The distribution of last-layer hidden states in the LLM is entirely inconsistent with that of the token embeddings.* As shown in Figure 2 and Figure 7, the distributions of hidden states and token embeddings after PCA dimensionality reduction differ substantially. Statistically, their

means, variances, and inter-distribution distances are all large. From a modeling perspective, shallow parameters (e.g., RMSNorm) encountered during pre-training are exposed only to the token embedding distribution. Feeding vectors from an unfamiliar distribution into the model can thus induce distribution shift or other unpredictable effects, which is one of the reasons why methods such as COCONUT perform poorly.

## 3.2 Non-Full Effective Rank of Token Embeddings in Large Language Models

Building on Observation 1, we posit that the latent token representations should ideally align with the distributional geometry of the token embeddings. To examine this property, we analyze the singular value spectra of token embedding matrices across different LLMs.

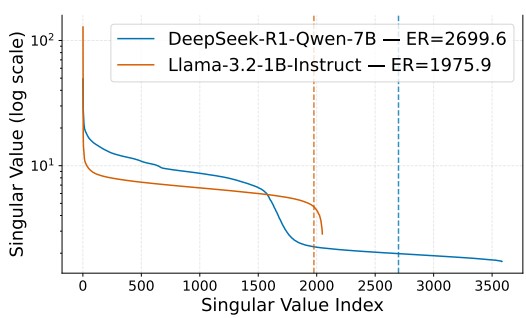

**Observation 2.** *The Effective Rank of LLM Token Embeddings Is Not Full.* The resulting curves in Figure 3 exhibit a rapid decay rather than a flat plateau, indicating that only a subset of singular directions contributes substantially to the embedding space. The effective rank of word embeddings is far below the theoretical maximum, indicating that although the embedding matrix has high nominal dimensionality, it occupies only a low-dimensional subspace.

**Figure 3:** Decay curves of singular values for embedding matrices in various LLMs. The legend indicates the effective rank of each embedding matrix.

Thus, the semantic space of LLMs can be regarded as inherently low-dimensional.

These findings suggest that latent tokens should not be treated as unconstrained hidden vectors, but rather as structured elements residing within the embedding subspace. Guided by this insight, we formally define latent tokens.

**Definition 1** (*Soft Embedding*). Formally, a latent token $z \in \mathbb{R}^d$ can be expressed as a linear combination of token embeddings:

$$z = \sum_{i=1}^{V} \alpha_i e_i, \quad \text{with } \alpha_i \in \mathbb{R}, e_i \in \mathbb{R}^d, \tag{1}$$

where $\{e_i\}_{i=1}^{V}$ denotes the embedding vectors of the vocabulary, and the coefficients $\{\alpha_i\}$ determine the semantic mixture. This formulation ensures that latent tokens remain aligned with the intrinsic semantic geometry of the LLM's vocabulary space, providing a principled interpretation of latent tokens as interpolation within the embedding manifold.

## 4 Latent-SFT: Equipping Large language Models with Latent Reasoning

With **Definition 1**, we introduce Latent-SFT to bring latent reasoning into LLMs. As a prerequisite, latent tokens must be generated (Definition 1), since they provide the labels needed for supervision. Thereafter, the model is trained, in analogy to explicit reasoning, to generate these latent tokens and perform latent reasoning. The overall training framework is illustrated in Figure 4.

### 4.1 Generating Latent Tokens via Induction-Supervision Masking

Latent tokens are designed to be: (1) **semantic compactness**, replacing multiple explicit tokens; (2) **semantic compatibility**, constrained to the semantic space of explicit tokens (Section 3); and (3) **semantic correctness**, capable of producing the right answer. We implement these properties within an encoder–decoder framework through three key steps: information compression, representation alignment, and supervised decoding.

**Information Compression Process (Semantic Compactness).** We insert special tokens into the explicit reasoning sequence to segment and capture intermediate reasoning information. Given an

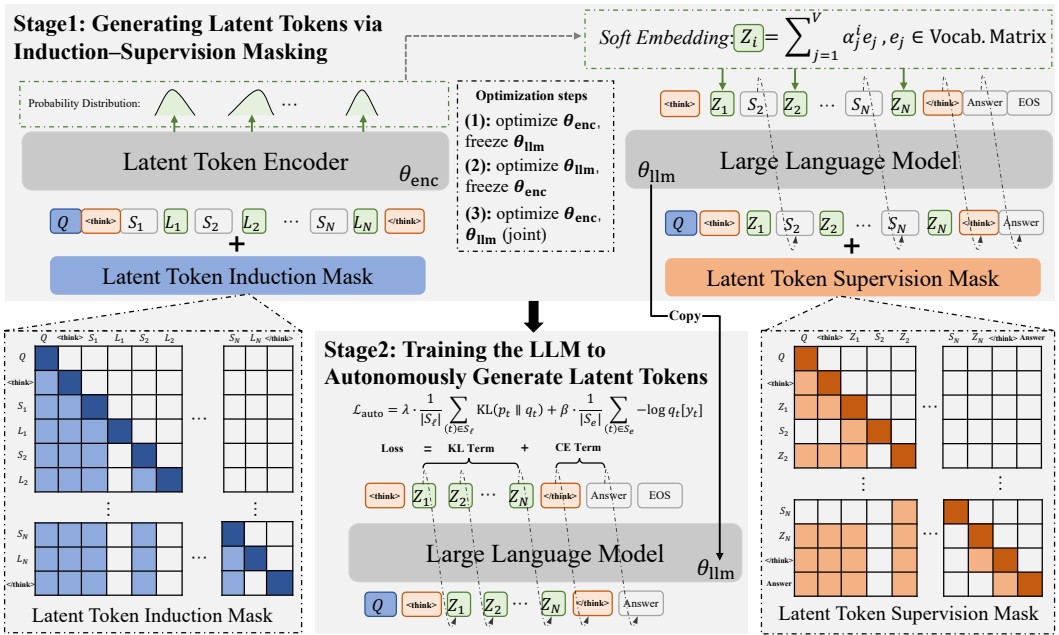

**Figure 4:** Overview of Latent-SFT. The training process consists of two phases: (a) Generating Latent Tokens via Induction–Supervision Masking. In the mask sub-figure, components such as $Q$, $S_i$ and Answer (which may span multiple tokens) are implemented via a standard autoregressive attention mechanism. The Latent Token Encoder shares both structure and initialization with the base LLM. (b) Training the LLM to Autonomously Generate Latent Tokens. In this phase, the LLM is trained to perform latent reasoning independently, using a weighted combination of KL loss and CE loss.

input $X = [Q, <\text{think}>, T, </\text{think}>]$, we define a segmentation function $\text{Seg}(\cdot)$ that divides $T$ (explicit reasoning sequence) into $N$ subsegments, i.e., $T = \{S_i\}_{i=1}^N$. This segmentation can be performed either by a fixed token length (compression ratio) $r$ or by grouping semantically continuous thoughts. After each $S_i$, we insert a special token $L_i$ and apply the Latent Token Induction Mask (LTIM) to restrict each $L_i$ to attend only to the explicit reasoning information preceding it (see Figure 4). This design ensures that the semantic content encoded by latent tokens grows with reasoning completeness, facilitating subsequent model learning.

**Representation Alignment Process (Semantic Compatibility).** Given the inserted special tokens $\{L_i\}_{i=1}^N$, we first obtain their hidden states $h_i \in \mathbb{R}$ from the (LTIM-masked) model. To align each latent token with the LLM's vocabulary manifold, we project $h_i$ onto the vocabulary simplex and then reconstruct its embedding as a mixture of token embeddings, in accordance with Definition 1. Concretely, let $E = [e_1, ..., e_V] \in \mathbb{R}^{d \times V}$ denote the (input) embedding matrix and $W \in \mathbb{R}^{d \times V}$ the LM head (often $W = E$) under weight tying. We compute vocabulary logits and a probability vector

$$\boldsymbol{\ell}_i = W^\top \mathbf{h}_i, \qquad \boldsymbol{\alpha}_i = \text{softmax}(\boldsymbol{\ell}_i/T) \in \Delta^{V-1}, \tag{2}$$

where $T > 0$ is an optional temperature. The representation of the $i$-th latent token is then obtained by linear interpolation within the embedding space:

$$\mathbf{z}_i = E\boldsymbol{\alpha}_i = \sum_{v=1}^V \alpha_{i,v}\, \mathbf{e}_v. \tag{3}$$

which instantiates Definition 1 (*Soft Embedding*) with coefficients given by the model-implied vocabulary distribution. This construction guarantees $z_i$ lies in the span (and, with softmax, the convex hull) of $\{e_v\}_{v=1}^V$, thereby enforcing **semantic compatibility**: latent tokens remain confined to the intrinsic geometry of the LLM's vocabulary space. In training Stage 2, $\alpha_i$ serves as the supervision signal for latent positions (KL term), while explicit positions are optimized with CE; optionally, one may prune $\alpha_i$ to top-$k$ and renormalize to reduce computation without departing from the embedding manifold.

**Supervised Decoding Process (Semantic Correctness).** Given the latent representations $\{z_i\}_{i=1}^N$ from the alignment step, we use each $z_i$ to decode the content from the $(i+1)$-th explicit reasoning subsegment through to the final answer. Let the prefix up to step $i$ be $\Pi_i = [Q, <\text{think}>, z_1, S_2, z_2, ..., z_i]$. Under the Latent Token Supervision Mask (LTSuM), tokens in $\mathcal{Y}_i = [S_{i+1}, ..., S_N, </\text{think}>, \text{Answer}]$ are decoded conditioned only on $\Pi_i$ (in particular on $[Q, <\text{think}>, z_1, ..., z_i]$), while attention to future latent tokens and to past explicit reasoning steps is blocked. Denoting by $\mathcal{J}_i$ the index set of tokens in $\mathcal{Y}_i$, the supervised decoding objective is

$$\mathcal{L}_{\text{sup}} \ = \ \frac{1}{N} \sum_{i=1}^N \frac{1}{|\mathcal{J}_i|} \sum_{t \in \mathcal{J}_i} \Big( -\log p_\theta\big(x_t \,\big|\, \Pi_i; \text{LTSuM}\big)\Big). \tag{4}$$

**Training Schedule.** In practice, the strong masking constraints (LTIM/LTSuM) make naive SFT optimization difficult. We therefore adopt an EM-style alternating schedule. Let the complete set of parameters be $\theta = (\theta_{\text{enc}}, \theta_{\text{llm}})$. First, with the LLM frozen, we optimize the encoder only, $\theta_{\text{enc}}^{t+1} = \arg\ \min_{\theta_{\text{enc}}} \mathcal{L}_{\text{sup}}(\theta_{\text{enc}}, \theta_{\text{llm}}^t)$. Next, with the encoder frozen, we optimize the LLM, $\theta_{\text{llm}}^{t+1} = \arg\ \min_{\theta_{\text{llm}}} \mathcal{L}_{\text{sup}}(\theta_{\text{enc}}^{t+1}, \theta_{\text{llm}})$. Finally, we jointly fine-tune both components by minimizing, $(\theta_{\text{enc}}, \theta_{\text{llm}}) \leftarrow \arg\ \min_{\theta_{\text{llm}}, \theta_{\text{enc}}} \mathcal{L}_{\text{sup}}(\theta_{\text{enc}}, \theta_{\text{llm}})$. This alternating–joint procedure stabilizes training under the constraints while steadily improving convergence.

## 4.2 TRAINING THE LLM TO AUTONOMOUSLY GENERATE LATENT TOKENS

After generating latent tokens with the three desired characteristics, we concatenate them with the explicit tokens. The resulting sequence $X = [Q, <\text{think}>, z_1, ..., z_n, </\text{think}>, \text{Answer}]$ then serves as the input for Stage 2 (in Figure 4(b)). Denote the index sets of latent and explicit positions by $S_{\text{lat}}, S_{\text{exp}}$, respectively. We use $p_t = \alpha_t$ as the soft label for the latent position, and ground-truth tokens $y_t$ for explicit positions. Let $q_t = \text{softmax}(W^T h_t)$ be the decoder's predictive distribution at position $t$. We optimize the decoder with a KL term on latent slots and a CE term on explicit slots:

$$\mathcal{L}_{\text{auto}}(\theta_{\text{llm}}) = \lambda \cdot \frac{1}{|S_{\text{lat}}|} \sum_{t \in S_{\text{lat}}} \text{KL}\big(p_t \,|\, q_t\big) \ + \ \beta \cdot \frac{1}{|S_{\text{exp}}|} \sum_{t \in S_{\text{exp}}} \big( -\log q_t[\,y_t\,]\big). \tag{5}$$

Optionally, a temperature $T > 0$ can be applied to both teacher and student distributions (with the usual $T^2$ scaling) and $p_t$ may be pruned to top-$k$ then renormalized. Minimizing $\mathcal{L}_{\text{auto}}$ trains the decoder to autonomously generate latent token sequences $[z_1, ..., z_N]$ along with the final answer.

## 5 EXPERIMENTS

### 5.1 EXPERIMENTAL SETUP

**Datasets.** To comprehensively assess the effectiveness of our method, we conducted experiments on both low-difficulty mathematical reasoning datasets (GSM8k-Aug (Deng et al., 2024), GSM-Hard (Gao et al., 2023), SVAMP (Patel et al., 2021), and MultiArith (Roy & Roth, 2015)) and high-difficulty datasets (Math500 (Hendrycks et al., 2021) and AIME24). For the low-difficulty tasks, we trained on the GSM8k-Aug training split to ensure fair comparison with prior work. For the high-difficulty tasks, due to limited computational resources, we used a subset of Open-R1 (Hugging Face, 2025) containing explicit reasoning chains shorter than 4k tokens, resulting in 57,544 training samples. This restriction inevitably constrained the performance of our method.

**Baselines and Models.** The baselines considered in our experiments include COT-SFT (Wei et al., 2022), iCOT (Deng et al., 2024), COCONUT (Hao et al., 2024), CODI (reproduced) (Tan et al., 2025), and CoLaR (Tan et al., 2025). For low-difficulty tasks, we adopt LLaMA-3.2-1B-Instruct (Dubey et al., 2024), while for high-difficulty tasks, we employ DeepSeek-Distill-Qwen-7B (DeepSeek-AI et al., 2025).

**Evaluation Metrics and Details.** Our evaluation measures both accuracy (Pass@N) and efficiency (# L, the number of tokens in the reasoning chain). For the low-difficulty datasets, we conducted five runs with different random seeds. For the high-difficulty datasets, Math500 was

**Table 1:** Experimental results on four low-difficulty datasets. The number in brackets represents the compression ratio $r$. "⬛" marks the best result, and "⬛" marks the second best. - Hidden State refers to removing the Representation Alignment step and directly using the last-layer hidden state as the representation of latent tokens. - w/o LTIM and - w/o LTSuM denote removing the Latent Token Induction Mask and Latent Token Supervision Mask, respectively, reducing the model to a standard autoregressive attention mechanism.

| | GSM8k-Aug | | GSM-Hard | | SVAMP | | MultiArith | | Average | | |
| --- | --- | --- | --- | --- | --- | --- | --- | --- | --- | --- | --- |
| | Pass@1 | # L | Pass@1 | # L | Pass@1 | # L | Pass@1 | # L | Pass@1 | # L | Pass@1/# L |
| CoT-SFT | 49.4$_{\pm.7}$ | 25.6$_{\pm.1}$ | **11.9**$_{\pm.2}$ | 34.2$_{\pm.1}$ | **59.8**$_{\pm.3}$ | 12.1$_{\pm.0}$ | 93.2$_{\pm.5}$ | 13.7$_{\pm.1}$ | **53.6** | 21.4 | 2.50 |
| iCoT | 19.8$_{\pm.2}$ | 0.00$_{\pm.0}$ | 3.87$_{\pm.2}$ | 0.00$_{\pm.0}$ | 36.4$_{\pm.5}$ | 0.00$_{\pm.0}$ | 38.2$_{\pm.7}$ | 0.00$_{\pm.0}$ | 24.6 | 0.00 | - |
| COCONUT | 23.1$_{\pm.3}$ | 6.00$_{\pm.0}$ | 5.49$_{\pm.3}$ | 6.00$_{\pm.0}$ | 40.7$_{\pm.7}$ | 6.00$_{\pm.0}$ | 41.1$_{\pm.2}$ | 6.00$_{\pm.0}$ | 27.6 | 6.00 | 4.60 |
| CODI | 13.3$_{\pm.6}$ | 6.00$_{\pm.0}$ | 2.97$_{\pm.2}$ | 6.00$_{\pm.0}$ | 21.7$_{\pm.7}$ | 6.00$_{\pm.0}$ | 19.2$_{\pm.8}$ | 6.00$_{\pm.0}$ | 14.3 | 6.00 | 2.38 |
| CoLaR-5 | 26.8$_{\pm.2}$ | 5.57$_{\pm.0}$ | 5.87$_{\pm.1}$ | 6.53$_{\pm.0}$ | 48.4$_{\pm.5}$ | 2.95$_{\pm.0}$ | 86.4$_{\pm.4}$ | 3.21$_{\pm.0}$ | 41.7 | 4.57 | 9.12 |
| CoLaR-2 | 40.1$_{\pm.2}$ | 12.7$_{\pm.0}$ | 9.08$_{\pm.0}$ | 14.0$_{\pm.0}$ | 54.9$_{\pm.2}$ | 6.11$_{\pm.0}$ | 91.3$_{\pm.1}$ | 7.35$_{\pm.0}$ | 48.8 | 10.0 | 4.88 |
| Latent-SFT(4) | 44.6$_{\pm.0}$ | 6.64$_{\pm.0}$ | 10.2$_{\pm.0}$ | 7.68$_{\pm.0}$ | 55.6$_{\pm.1}$ | 3.33$_{\pm.0}$ | 93.6$_{\pm.0}$ | 4.11$_{\pm.0}$ | 51.0 | 5.44 | **9.37** |
| - Hidden State | 36.7$_{\pm.4}$ | 6.61$_{\pm.1}$ | 8.41$_{\pm.2}$ | 7.77$_{\pm.0}$ | 44.4$_{\pm.2}$ | 3.22$_{\pm.0}$ | 90.3$_{\pm.1}$ | 4.06$_{\pm.0}$ | 45.0 | 5.41 | 8.32 |
| - w/o LTIM | 38.8$_{\pm.1}$ | 6.59$_{\pm.1}$ | 8.67$_{\pm.1}$ | 8.01$_{\pm.1}$ | 48.9$_{\pm.1}$ | 3.34$_{\pm.1}$ | 90.8$_{\pm.0}$ | 4.36$_{\pm.0}$ | 46.8 | 5.58 | 8.39 |
| - w/o LTSuM | 42.6$_{\pm.0}$ | 6.70$_{\pm.0}$ | 9.98$_{\pm.1}$ | 7.60$_{\pm.0}$ | 53.9$_{\pm.1}$ | 3.36$_{\pm.0}$ | 92.6$_{\pm.0}$ | 4.27$_{\pm.0}$ | 49.7 | 5.48 | 9.07 |
| Latent-SFT(2) | **50.4**$_{\pm.1}$ | 12.4$_{\pm.0}$ | 10.9$_{\pm.0}$ | 13.9$_{\pm.0}$ | 57.8$_{\pm.1}$ | 5.98$_{\pm.0}$ | **93.8**$_{\pm.0}$ | 7.21$_{\pm.0}$ | 53.2 | 9.87 | 5.39 |
| - Hidden State | 41.1$_{\pm.3}$ | 12.8$_{\pm.0}$ | 9.01$_{\pm.1}$ | 14.2$_{\pm.1}$ | 49.6$_{\pm.2}$ | 6.17$_{\pm.0}$ | 91.1$_{\pm.0}$ | 7.29$_{\pm.0}$ | 47.7 | 10.1 | 4.72 |
| - w/o LTIM | 41.3$_{\pm.1}$ | 12.9$_{\pm.1}$ | 9.32$_{\pm.0}$ | 14.0$_{\pm.0}$ | 53.6$_{\pm.1}$ | 6.25$_{\pm.1}$ | 91.5$_{\pm.0}$ | 7.96$_{\pm.0}$ | 48.9 | 10.3 | 4.75 |
| - w/o LTSuM | 49.0$_{\pm.0}$ | 12.1$_{\pm.0}$ | 10.3$_{\pm.0}$ | 13.7$_{\pm.0}$ | 56.5$_{\pm.2}$ | 6.22$_{\pm.0}$ | 93.0$_{\pm.0}$ | 7.23$_{\pm.0}$ | 52.2 | 9.81 | 5.32 |

**Table 2:** Experimental results on high-difficulty datasets. We primarily examine the performance gap between *Soft Embedding* and hidden state representations. Long inference chains amplify error accumulation in latent reasoning, while training restricted to lengths of 4k further limits performance. Consequently, the accuracy drop on AIME24 is acceptable. Since prior latent baselines were neither trained nor evaluated on high-difficulty datasets, their results are not reported.

| Model | MATH-500 | | | AIME24 | | |
| --- | --- | --- | --- | --- | --- | --- |
| | Pass@1 | # L | Pass@1/# L(100×) | Pass@1 | # L | Pass@1/# L(100×) |
| COT-SFT | 92.6$_{\pm1.19}$ | 3616$_{\pm232}$ | 2.56 | 54.4$_{\pm1.74}$ | 12992$_{\pm346}$ | 0.42 |
| COT-SFT (1k-Budget) | 75.5$_{\pm1.38}$ | 1017$_{\pm6.44}$ | 7.42 | 12.5$_{\pm1.28}$ | 1046$_{\pm0.01}$ | 1.19 |
| COT-SFT (2k-Budget) | 85.8$_{\pm1.83}$ | 1670$_{\pm19.1}$ | 5.14 | 23.7$_{\pm1.27}$ | 2066$_{\pm1.08}$ | 1.15 |
| COT-SFT (3k-Budget) | 89.2$_{\pm0.90}$ | 2057$_{\pm26.8}$ | 4.34 | 33.5$_{\pm1.40}$ | 3017$_{\pm8.06}$ | 1.11 |
| COT-SFT (4k-Budget) | 90.5$_{\pm0.66}$ | 2318$_{\pm40.8}$ | 3.90 | 39.0$_{\pm1.15}$ | 4060$_{\pm13.7}$ | 0.96 |
| COT-SFT (NoThink) | 77.5$_{\pm1.88}$ | 938.5$_{\pm131}$ | 8.26 | 22.3$_{\pm1.13}$ | 4382$_{\pm234}$ | 0.51 |
| Latent-SFT (Hidden State) | 67.8$_{\pm0.94}$ | 698.1$_{\pm22.1}$ | 9.71 | 7.78$_{\pm2.08}$ | 1509$_{\pm114}$ | 0.51 |
| Latent-SFT (*Soft Embedding*) | 79.8$_{\pm1.35}$ (12.0% ↑) | 759.3$_{\pm36.9}$ | **10.51** | 19.2$_{\pm1.60}$ (11.4% ↑) | 1484$_{\pm420}$ | **1.29** |

evaluated four times, while AIME24 was evaluated 64 times. We report Pass@1 and # L as the mean ± 95% confidence interval (CI). To jointly assess accuracy and efficiency, we also report their ratio, Pass@1/# L. Additional implementation details are provided in Appendix B.

## 5.2 MAIN RESULTS

**Performance on Low-Difficulty Tasks.** Table 1 compares Latent-SFT with state-of-the-art baselines under different compressed ratio $r$ settings. Relative to the latent reasoning baseline, Latent-SFT consistently achieves higher accuracy, shorter reasoning chains, and substantial gains in the Pass@1 metric. These results confirm that defining latent tokens as linear combinations of the vocabulary matrix—thus aligning them with the same semantic space as explicit tokens—is more effective than the traditional last-hidden-state definition. Furthermore, Latent-SFT demonstrates strong out-of-distribution generalization, achieving state-of-the-art performance on all three out-of-distribution datasets and even surpassing COT-SFT on MultiArith. This advantage arises from our latent token definition, which is more compatible with explicit tokens and thus better leverages token representations learned during pre-training, leading to improved generalization.

It is noteworthy that Latent-SFT(2) surpasses explicit COT-SFT in Pass@1 on both GSM8k-Aug and MultiArith, while reducing inference chain length by roughly one quarter on average. This finding demonstrates that explicit reasoning does not represent the upper bound of latent reasoning methods. With an appropriate definition and learning strategy for latent tokens, latent reasoning can exceed explicit reasoning models in accuracy. When considering both Pass@1 and #L, Latent-SFT delivers nearly a fourfold improvement over traditional COT-SFT, demonstrating strong overall performance. The real inference example is provided in the Figure 10.

**Performance on High-Difficulty Reasoning Tasks.** Experimental results on high-difficulty mathematical tasks are shown in Table 2. We found that the accuracy of Latent-SFT on the AIME24

dataset dropped sharply compared to the COT-SFT model. We speculate that this is primarily due to **the increased difficulty of learning latent reasoning models with long reasoning chains**. Simply put, as reasoning chains length increases, it becomes more difficult to learn the three features required for latent tokens, resulting in a deterioration in latent token representation. This deterioration in latent token representation leads to cumulative errors in the latent reasoning stage, ultimately resulting in a drop in accuracy. Furthermore, due to training resource limitations, our model's **training examples have reasoning chains no longer than 4k in length**, further hindering its generalization to the AIME24 dataset, which requires an average of 12k tokens. (As evidence, our model performs significantly better on the MATH500 dataset, which requires shorter reasoning chains, than on AIME24, which requires longer chains.) Therefore, adapting latent reasoning to high-difficulty mathematical reasoning datasets remains a challenging task.

In terms of inference chain length, Latent-SFT remains effective, achieving substantial reductions on both datasets. Under the same inference chain length, our method achieves the highest accuracy. In terms of inference efficiency, our approach outperforms various explicit reasoning methods, achieving the best overall efficiency. More importantly, *Soft Embedding*, the core definition proposed in this work, consistently outperforms Hidden State even on high-difficulty datasets, thereby validating the generalizability of **Definition 1**.

## 5.3 Ablation Study

To comprehensively assess the effectiveness of Latent-SFT, we perform an ablation study focusing on its three core components for latent token generation: LTIM, LTSuM, and *Soft Embedding*.

**Replacing LTIM and LTSuM with Standard Autoregressive Attention.** LTIM is employed during the information compression process to restrict each special token to attend only to preceding explicit reasoning steps, while preventing attention among special tokens themselves. When replaced with standard autoregressive attention, the hidden states of later special tokens may be influenced by earlier ones. In the early stages of training, before convergence, this effect can be particularly detrimental. As shown in Table 1, across Latent-SFT models with different $r$ values, removing LTIM leads to a substantial drop in Pass@1 (about 5% on average), underscoring the necessity of LTIM.

LTSuM is applied in the supervised decoding process to ensure that each latent token can decode the subsequent explicit reasoning steps, thereby maintaining semantic correctness. Replacing it with standard autoregressive attention—i.e., removing all explicit reasoning sub-segments and allowing only latent tokens to generate the final answer—eliminates intermediate supervision during latent token learning and may cause semantic entanglement. As shown in Table 1, removing LTSuM leads to a substantial drop in accuracy, demonstrating its necessity.

**Replace Latent Token Representation from *Soft Embedding* to the Last Hidden State** Soft Embedding serves as our core definition of latent tokens. While Table 1 shows that it already outperforms traditional last-hidden-state methods such as COCONUT and CODI, for a fair comparison we replace Soft Embedding with the last hidden state in our framework while keeping all other settings unchanged. As reported in Tables 1 and Table 2, Soft Embedding achieves substantially better performance than the last hidden state on both low- and high-difficulty mathematical reasoning tasks, thereby empirically validating its effectiveness.

## 6 Analysis: Understanding the Essence of Latent Reasoning

In order to probe the fundamental nature of latent reasoning, we raise a central question: *What, precisely, does latent reasoning capture during the reasoning process?* Inspired by COCONUT's perspective that latent reasoning explores multiple reasoning paths simultaneously, we propose the following two hypotheses.

**Hypothesis 1. Latent Reasoning Represents the Compression of a Single Reasoning Path.** To quantify how much explicit information is preserved at each step of latent reasoning, we introduce the **Top-K Effective Compression Ratio (ECR@K)**. This metric measures the proportion of

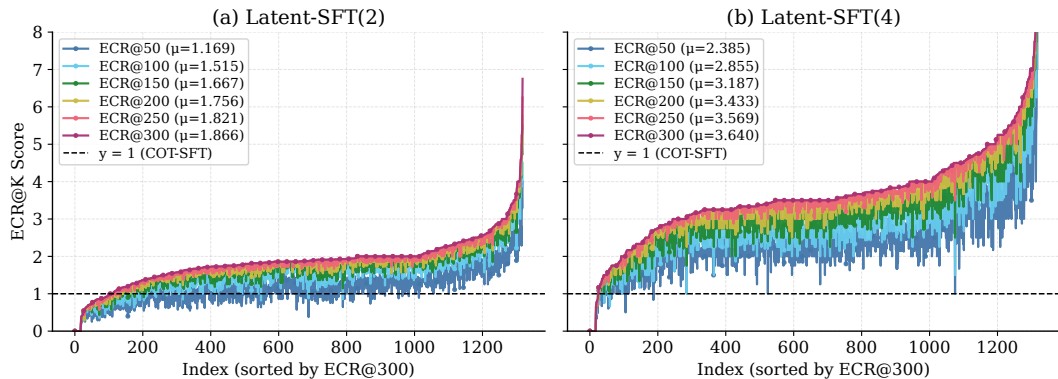

**Figure 5:** Latent-SFT's ECR@K values on GSM8k-Aug test samples. Each curve represents the per-sample ECR@K, with samples sorted along the x-axis by their ECR@300 values.

explicit chain tokens that fall within the Top-K of the probability distribution at the corresponding latent step. Larger values indicate stronger horizontal compression across the reasoning chain, while values greater than one reflect true compression of the explicit chain. The formal definition is provided in the Appendix D.2.

The experimental results are presented in Figure 5. Across models trained with different compression ratios r and varying K values, the ECR scores exceed 1 for almost all samples. This quantitatively validates our hypothesis that latent reasoning compresses a single reasoning path. From the training perspective, the model indeed performs information compression along a single reasoning path. Moreover, comparing Figure 5(a) and (b), doubling the training compression ratio nearly doubles the ECR score during inference, further demonstrating the effectiveness of our training strategy.

**Hypothesis 2. Latent Reasoning Represents the Superposition of Multiple Reasoning Paths.** To examine whether latent reasoning corresponds to the superposition of multiple paths, we define the effective global parallelism ($N_{eff}$), which quantifies global evidence for the presence of multiple parallel reasoning paths. A larger value reflects a higher degree of parallelism. In addition, we introduce the Top-2 Score, defined as the ratio of the probabilities of the two most likely paths. Formal definitions are provided in the Appendix D.3. To assess the parallel reasoning capability of latent models, we construct multiple alternative reasoning chains corresponding to a single explicit reasoning path within GSM8k-Aug dataset (Multi-Chain GSM8k-Aug dataset in Appendix C).

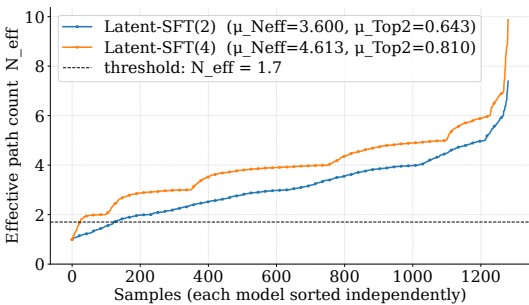

**Figure 6:** $N_{eff}$ values of Latent-SFT on the Multi-Chain GSM8k-Aug dataset. The legend reports the mean values of both $N_{eff}$ and the Top-2 Score.

Figure 6 presents the analysis results. Across models with different compression ratios, most samples exceed the conservative threshold for significant parallelism (1.7), indicating that latent reasoning reflects the superposition of multiple reasoning chains. On average, evidence is distributed across roughly 3–4 reasoning paths rather than concentrated on a single path. The average Top-2 ratio exceeds 0.6, suggesting strong "two-way competition" and parallel reasoning. Furthermore, as the compression ratio increases, the degree of parallelism also rises. Case studies are provided in the Figure 9.

## 7 CONCLUSION

In this work, we propose Latent-SFT, a method that enables latent reasoning in LLMs by defining latent tokens within the column space of the vocabulary. Experimental results show that on low-difficulty mathematical reasoning datasets, Latent-SFT achieves performance comparable to

explicit COT-SFT while significantly shortening reasoning chains. On high-difficulty datasets, it outperforms methods based on last hidden state representations. Furthermore, our analysis demonstrates that latent reasoning embodies both the compression of a single reasoning chain and the superposition of multiple chains.

## 8 Ethics statement and Reprodicibility statement

This work does not raise ethical concerns. The code, datasets, and models of our method are available in an anonymous GitHub repository https://anonymous.4open.science/r/Latent-SFT-CF17.

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

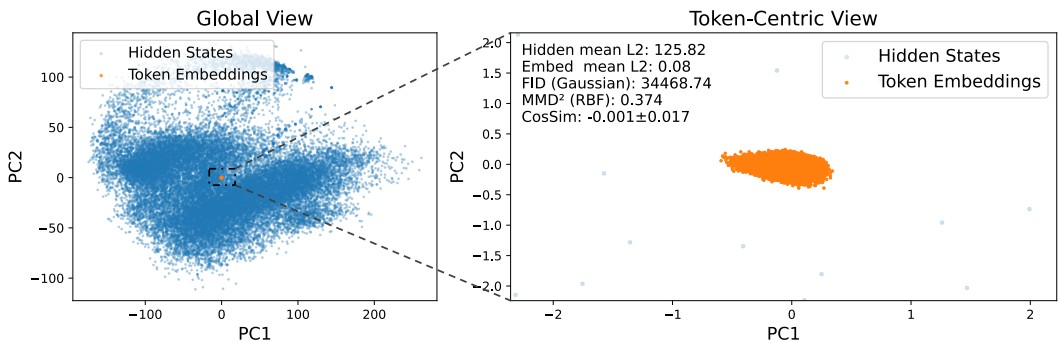

**Figure 7:** Visualization of last-layer hidden states and token embeddings of the Deepseek-Distill-Qwen-7B model on the Math500 dataset.

## A    USE OF LLMs

We employ GPT-5 to polish the language of our manuscripts. Specifically, we first draft the text, then use GPT-5 to refine it into a more coherent form. After verifying its logical and semantic consistency, we incorporate the polished version into the main text.

## B    IMPLEMENTATION DETAILS

Since Latent-SFT requires weighting with respect to the vocabulary matrix, we freeze the vocabulary parameters during training. All models are trained using LoRA with rank 64. Following CO-CONUT, we initialize all methods from the COT-SFT model to accelerate convergence. In practice, we find that a fixed-length segmentation function performs better than a semantic-based segmentation. Therefore, all results reported in this paper are based on fixed-length segmentation functions. For LLaMA-3.2-1B-Instruct, we set the learning rate to $1 \times 10^{-4}$ in Step 1 and $4 \times 10^{-4}$ in Step 2. For DeepSeek-Distill-Qwen-7B, the learning rates are $5 \times 10^{-5}$ (Step 1) and $1 \times 10^{-5}$ (Step 2). Following DeepSeek-AI et al. (2025), we use a decoding temperature of 0.6 and top-p of 0.95. We did not perform parameter tuning for $\lambda$ and $\beta$; setting both to 1 yielded satisfactory performance.

During evaluation, low-difficulty mathematical reasoning datasets were inferred using the Transformers library with a maximum sequence length of 128. High-difficulty datasets were evaluated with the improved SGlang library, supporting a maximum token length of 32,768. All training and inference experiments were conducted on 8×A100 or 8×H100 GPUs.

## C    CONSTRUCTION OF THE MULTI-CHAIN GSM8K-AUG DATASET

We carefully designed prompts and employed the advanced GPT-5 model to generate preliminary candidate reasoning chains. Given the simplicity and limited number of questions, we manually inspected and removed examples that either failed to follow the reasoning format or contained incorrect logic. To further maximize diversity, we retained only reasoning chains with pairwise similarity below 0.9 based on edit distance. An example prompt call is illustrated in Figure 8. Among the 1,319 data points, 1,310 contained multiple reasoning paths.

## D    ANALYTICAL METRICS

### D.1    NOTATION AND PRELIMINARIES

Let $\mathcal{V}$ be the vocabulary, $|\mathcal{V}| = V$. For a given problem, fix a reference explicit chain (from an explicit CoT solution)

$$\boldsymbol{x} = (x_1, \ldots, x_{L_{\exp}}), \quad x_i \in \mathcal{V}. \tag{6}$$

**Instruction:**
"Given a GSM8k problem, solution, and answer, identify alternative valid reasoning paths. Enclose each path within \\boxed{}. The paths must not contain any natural language, and should follow the same style as the provided example. Ensure the length of the solution and the number of reasoning steps <<>> remain consistent with the standard solution. Each reasoning path must strictly adhere to the problem's logical structure—do not generate unrelated or fabricated paths. If no alternative reasoning path exists, output \\boxed{False}.

{"problem": "Janet's ducks lay 16 eggs per day. She eats three for breakfast every morning and bakes muffins for her friends every day with four. She sells the remainder at the farmers' market daily for \$2 per fresh duck egg. How much in dollars does she make every day at the farmers' market?", "solution": "<<16–3–4=9>> <<9*2=18>>", "answer": "18"} "

**Response:**
"\boxed{<<<<16–(3+4)=9>> <<9*2=18>>}
\boxed{<<3+4=7>> <<(16–7)*2=18>>}     ✗ Rejected (edit-distance criterion against the solution violated).
\boxed{<<16–3=13>> <<(13–4)*2=18>>}
\boxed{<<16–4=12>> <<(12–3)*2=18>>}     ✗ Rejected (edit-distance criterion against the previous item violated).
\boxed{<<2*16=32>> <<32–2*3–2*4=18>>}"

**Figure 8:** An example of calling GPT5 and post-processing.

A latent model generates a latent reasoning chain consisting of $T$ latent steps, each with a vocabulary distribution

$$p_t \in \Delta^{V-1} \quad (t = 1, \ldots, T), \tag{7}$$

where $\Delta^{V-1}$ is the probability simplex. To quantify the extent of explicit information captured by each latent step (whether from a single chain or multiple chains), we introduce:

**Assumption 1.** During training, if each latent token is constrained to receive information from a fixed number of explicit tokens, then at inference time the amount of explicit information (in terms of token granularity) captured by each latent step is uniform.

This assumption is reasonable, as the model is expected to behave consistently during training and inference. So we align the two sequences by a ratio $r \in \mathbb{N}$: every $r$ explicit tokens are grouped to one latent step. Define the number of aligned steps

$$T' = \min\big(T, \lceil L_{\exp}/r \rceil\big), \tag{8}$$

and the explicit token set aligned to latent step $t$:

$$S_t = \big\{ x_{(t-1)r+1}, \ldots, x_{\min(tr, L_{\exp})} \big\}, \qquad |S_t| \le r. \tag{9}$$

Let $\mathcal{T}_K(p_t) \subset \mathcal{V}$ be the set of the $K$ highest-probability tokens under $p_t$.

### D.2 ECR@K (EFFECTIVE COMPRESSION RATE)

To verify whether latent reasoning corresponds to the compression of a single chain, we examine, *on average, how many aligned explicit tokens from the reference chain appear within the Top-K probabilities at each latent step.* Thus, we define

$$\text{ECR@K} \; = \; \frac{1}{T'} \sum_{t=1}^{T'} \big| S_t \cap \mathcal{T}_K(p_t) \big| \; \in \; [0, r]. \tag{10}$$

A value greater than 1 indicates genuine compression of a single reasoning chain, meaning that each latent step, on average, covers more than one token from the reference explicit chain.

### D.3 $N_{\text{eff}}$ AND TOP-2 SCORE

Given $M$ candidate explicit chains $\{\boldsymbol{x}^{(m)}\}_{m=1}^{M}$ (with lengths $L_m$), we align each chain to the $T$ latent steps using the same ratio $r$ as above. Let the explicit tokens aligned to step $t$ on chain $m$ be $S_{m,t} \subseteq \mathcal{V}$ (at most $r$ tokens), and let $\mathcal{T}_K(p_t)$ be the Top-$K$ tokens under $p_t$. Define the per-step *mass* of chain $m$ as

$$\text{mass}_{m,t}^{(K)} \;=\; \sum_{x \in S_{m,t} \cap \mathcal{T}_K(p_t)} p_t[x], \tag{11}$$

and aggregate it across steps to obtain a sequence-level evidence score

$$\text{score}_m \;=\; \frac{1}{T'} \sum_{t=1}^{T'} \log\big(\text{mass}_{m,t}^{(K)} + \varepsilon\big), \tag{12}$$

where $T' = \min\big(T, \lceil L_m/r \rceil\big)$, $\varepsilon > 0$ is a small constant, and $K$ controls robustness (setting $K = V$ recovers the full-mass version). We convert $\{\text{score}_m\}$ to a posterior over paths via a temperatured softmax

$$P_m \;=\; \frac{\exp(\text{score}_m/\tau)}{\sum_{j=1}^{M} \exp(\text{score}_j/\tau)} \quad (\tau = 1). \tag{13}$$

The **effective number of supported paths** is

$$N_{\text{eff}} \;=\; \exp\Big(-\sum_{m=1}^{M} P_m \log P_m\Big), \tag{14}$$

and the **Top-2 score** is the ratio between the two largest posteriors,

$$\text{Top-2} \;=\; \frac{P_{(2)}}{P_{(1)}} \quad \text{with} \quad P_{(1)} \geq P_{(2)}. \tag{15}$$

Intuitively, a larger $\text{score}_m$ means the model consistently places high probability on chain $m$ along the sequence; a larger $N_{\text{eff}}$ and higher Top-2 indicate stronger parallel support across multiple chains rather than collapse to a single path.

When only a single path exists in parallel, $N_{\text{eff}} = 1$. However, since different paths may partially overlap in their prefixes, we adopt a conservative baseline of $N_{\text{eff}} = 1.7$ (corresponding to approximately 70% probability mass on path 1 and 30% on path 2). Values exceeding this threshold are taken as evidence of parallel inference. In addition, to mitigate the influence of tail noise, we set $K = 100$.

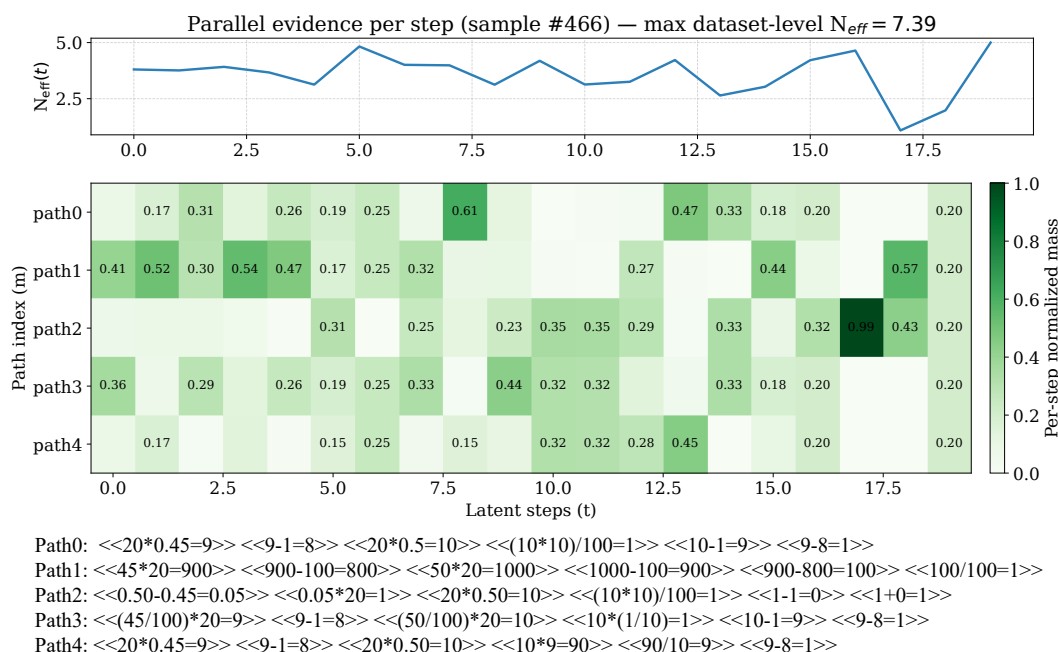

Path0: <<20*0.45=9>> <<9-1=8>> <<20*0.5=10>> <<(10*10)/100=1>> <<10-1=9>> <<9-8=1>>
Path1: <<45*20=900>> <<900-100=800>> <<50*20=1000>> <<1000-100=900>> <<900-800=100>> <<100/100=1>>
Path2: <<0.50-0.45=0.05>> <<0.05*20=1>> <<20*0.50=10>> <<(10*10)/100=1>> <<1-1=0>> <<1+0=1>>
Path3: <<(45/100)*20=9>> <<9-1=8>> <<(50/100)*20=10>> <<10*(1/10)=1>> <<10-1=9>> <<9-8=1>>
Path4: <<20*0.45=9>> <<9-1=8>> <<20*0.50=10>> <<10*9=90>> <<90/10=9>> <<9-8=1>>

**Figure 9:** An example of the maximum $N_{eff}$ value for the Latent-SFT(2) model on the Multi-Chain GSM8k-Aug dataset. At each step, the $N_{eff}$ value exceeds 1.7. Path0 denotes the original reasoning chain, while the remaining paths are additional candidate chains. The heatmap illustrates the fraction of paths supported at each latent step, clearly revealing the phenomenon of parallel reasoning.

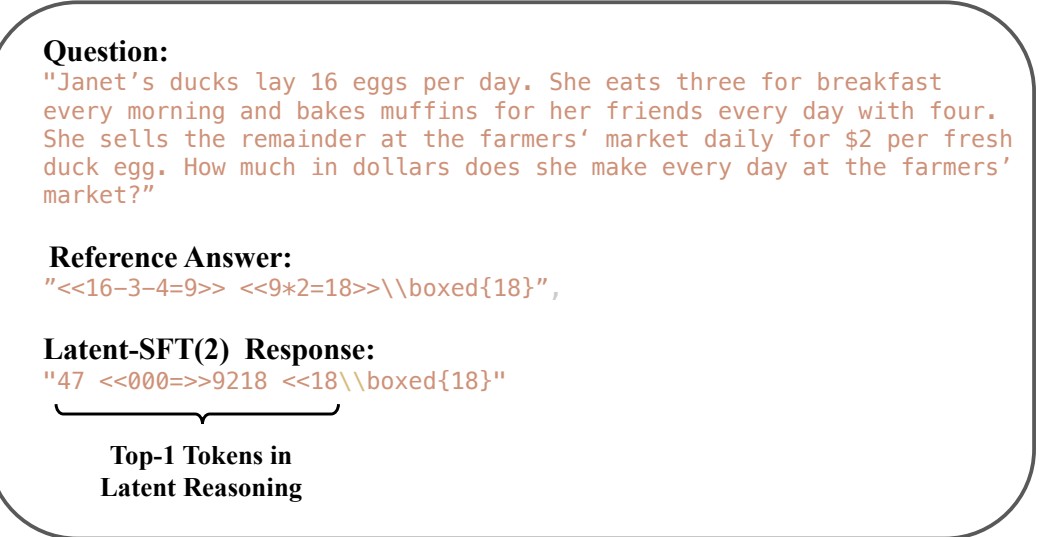

**Figure 10:** An example of Latent-SFT(2) reasoning on the GSM8k-Aug test set. The reasoning chain length is reduced by nearly half.

