# OpenReview forum: "Latent Reasoning in LLMs as a Vocabulary-Space Superposition"
_ICLR.cc/2026/Conference — Submitted to ICLR 2026_

### Official Review · Reviewer_uoyr · 2025-10-25

**Soundness:** 2
**Presentation:** 3
**Contribution:** 2
**Rating:** 4
**Confidence:** 3

**Summary:**

This paper introduces a novel framework for latent reasoning that compresses explicit chain-of-thought reasoning into a small number of latent tokens operating in the vocabulary embedding space. Empirically, the method achieves comparable or even better reasoning accuracy than explicit CoT while using 2× fewer tokens on low-difficulty tasks, but performance is still much lower than normal sft on difficult tasks.

**Strengths:**

* The paper is clearly written and easy to follow, with detailed experimental setups
* It proposes a complete and conceptually coherent system for compressing reasoning chains into latent tokens, unifying several ideas
* The evaluation includes both positive and negative results across multiple reasoning benchmarks, offering an honest and nuanced view of when latent reasoning helps and where it still struggles.

**Weaknesses:**

* The experimental design appears somewhat inconsistent: the authors evaluate low-difficulty tasks using LLaMA-3.2-1B-Instruct, while high-difficulty tasks are run on DeepSeek-Distill-Qwen-7B . Results on overlapping datasets or difficulty tiers using the same model would make the claims more convincing. (add low-diff tasks on deepseek and high-diff on llama)
* The separation of low- and high-difficulty evaluations raises questions about transferability of the learned reasoning-compression patterns. If the latent reasoning mechanism cannot generalize across tasks or difficulty levels, the approach’s practical benefit diminishes
* On challenging datasets such as Math500 and AIME24, performance lags behind normal sft. This weakens the practical significance of the method, as it seems most effective only for simpler reasoning tasks.

**Questions:**

see weakness

---

> ### Author Response · Authors · 2025-11-15
>
> Thank you for your thoughtful feedback on our work. I will now address your concerns point by point.
>
> > The experimental design appears somewhat inconsistent: the authors evaluate low-difficulty tasks using LLaMA-3.2-1B-Instruct, while high-difficulty tasks are run on DeepSeek-Distill-Qwen-7B . Results on overlapping datasets or difficulty tiers using the same model would make the claims more convincing. (add low-diff tasks on deepseek and high-diff on llama)
>
> Thank you very much for your suggestion. We did carefully consider this point. Based on empirical observations, high-parameter models often achieve near-perfect performance on low-difficulty datasets, while low-parameter models tend to perform poorly on high-difficulty benchmarks such as AIME24. Neither setting is ideal for conducting a meaningful, comparative analysis of our model against baselines. Therefore, we chose to evaluate low-parameter models on low-difficulty datasets, and high-parameter models on high-difficulty datasets, following the principle that appropriateness and fairness in evaluation are more important than uniformity.
>
> > The separation of low- and high-difficulty evaluations raises questions about transferability of the learned reasoning-compression patterns. If the latent reasoning mechanism cannot generalize across tasks or difficulty levels, the approach’s practical benefit diminishes
>
> It appears there may be a misunderstanding of our method. The evaluations on low- and high-difficulty tasks were conducted using different base models, but this does not imply a lack of transferability. On the contrary, our method is designed to be model-agnostic, and its successful application to models of different scales demonstrates its strong generalization capability.
>
> Furthermore, our experiments show that the method consistently improves inference efficiency across both low- and high-difficulty tasks, and across models of varying capacities. This highlights both the practical value and robust effectiveness of our approach.

---

> ### Author Response · Authors · 2025-11-15
>
> > On challenging datasets such as Math500 and AIME24, performance lags behind normal sft. This weakens the practical significance of the method, as it seems most effective only for simpler reasoning tasks.
>
> It’s possible that, due to a less-than-ideal presentation of our results, the accuracy metrics on high-difficulty mathematical reasoning datasets received more attention, while the inference chain length metric—which we believe is equally important—may have been overlooked. We would like to clarify the following key points:
>
> 1. Latent reasoning is a relatively new research paradigm, distinct from explicit reasoning. Most prior methods have struggled to match explicit reasoning even on low-difficulty benchmarks. Our work first closes that gap by nearly matching or surpassing explicit reasoning on low-difficulty datasets. We then extend the evaluation to more challenging benchmarks—Math500 and AIME24—to explore the comparative strengths and limitations of latent versus explicit reasoning. Notably, previous work rarely evaluated latent reasoning in such high-difficulty settings, whereas our study directly addresses this gap.
>
> 2. As shown in Table 2, although accuracy decreases on high-difficulty datasets, our method significantly reduces the required inference chain length.
>
> 3. On high-difficulty reasoning tasks, our approach outperforms traditional baselines in terms of accuracy. Specifically, we achieve **more than a 10% improvement** in accuracy on both the MATH500 and AIME24 benchmarks.
>
> 4. On low-difficulty datasets, our method achieves performance that is comparable to—or even surpasses—that of explicit reasoning. Compared to the previous state-of-the-art method CoLaR, our approach achieves **an improvement of over 10% in accuracy**.
>
> It's also important to highlight that our model was trained on sequences of only 4k tokens, which, as those familiar with reasoning tasks will understand, can heavily constrain model performance. To ensure a fair comparison, we evaluated the COT-SFT model with different reasoning-chain lengths and used a fair budget-forcing [1] procedure to obtain the answers. We also follow the evaluation settings NoThink [2] used in Qwen3 to ensure consistency in model assessment. To further unify the evaluation of accuracy and inference efficiency, we report the Pass@1 ⁄ #L (100×) metric, which reflects performance normalized by inference chain length. The experimental results are shown in the table below.

---

> ### Author Response · Authors · 2025-11-15
>
> | Model                         | MATH-500 Pass@1              | MATH-500 #L       | MATH-500 Pass@1⁄#(100×)      | AIME24 Pass@1                 | AIME24 #L       | AIME24 Pass@1⁄#(100×)        |
> |------------------------------|------------------------------|-------------------|-------------------|-------------------------------|-----------------|-----------------|
> | COT-SFT                      | 92.6 ± 1.19                  | 3616 ± 232        | 2.56       | 54.4 ± 1.74                   | 12992 ± 346     | 0.42       |
> | COT-SFT (1k-Budget)          | 75.5 ± 1.38               | 1017 ±6.44       | 7.42       | 12.5 ± 1.28                   | 1046 ± 0.01     | 1.19       |
> | COT-SFT (2k-Budget)          | 85.8 ± 1.83               | 1670 ±19.1       | 5.14       | 23.7 ± 1.27                   | 2066 ± 1.08     | 1.15      |
> | COT-SFT (3k-Budget)          | 89.2 ± 0.90               | 2057 ±26.8       | 4.34       | 33.5 ± 1.40                   | 3017 ± 8.06     | 1.11       |
> | COT-SFT (4k-Budget)          | 90.5 ± 0.66               | 2318 ±40.8       | 3.90       | 39.0 ± 1.15                   | 4060 ± 13.7     | 0.96       |
> | COT-SFT (NoThink)          | 77.5 ± 1.88              | 938.5 ±131       | 8.26       | 22.3 ± 1.13                   | 4382 ± 234     | 0.51       |
> | Latent-SFT (Hidden State)    | 67.8 ± 0.94                  | 698.1 ± 22.1      | 9.71       | 7.78 ± 2.08                   | 1509 ± 114      | 0.51       |
> | Latent-SFT (*Soft Embedding*)| 79.8 ± 1.35      | 759.3 ± 36.9      | **10.51**       | 19.2 ± 1.60         | 1484 ± 420      | **1.29**       |
>
> As shown in the table above, **under the same inference chain length, latent reasoning consistently outperforms explicit reasoning on both Math500 and AIME24**. Furthermore, across various baseline inference lengths, our model consistently achieved the highest inference efficiency, as measured by the Pass@1⁄#L (100×) metric. This setup ensures a fair and direct comparison between the two paradigms.
>
> In our experimental design, budget-forcing refers to constraining the model to reason within a specified length limit. To enforce this, we appended the instruction:
> ```text
> "Considering the limited time specified by the user, I will now provide the answer based on the above thinking.\n</think>\n\n"
> ```
> This prompt forces the model to stop reasoning at the desired point. In contrast, NoThink involves simply appending:
> ```text
> "\n</think>\n\n"
> ```
> to the prompt, allowing the model to generate the answer without a predefined stopping point.
>
> We would like to reiterate the key contributions of our work:
>
> 1. **From low- to high-difficulty mathematical reasoning:**
> Our model outperforms explicit reasoning on low-difficulty mathematical reasoning datasets and surpasses the previous state-of-the-art latent reasoning method **by over 10%**. We also provide **a comprehensive analysis** of the differences between latent and explicit reasoning on high-difficulty benchmarks, offering valuable insights for the community. Furthermore, we show that our method achieves higher inference efficiency than explicit reasoning in these challenging settings.
>
> 2. **A new definition of latent reasoning:**
> We extend the latent token representation from hidden states to soft embeddings based on preliminary experiments, and we show that soft embeddings consistently outperform hidden states on reasoning datasets of varying difficulty.
>
> 3. **Quantitative analysis of latent reasoning paths:**
> While COCONUT provides a qualitative observation that latent reasoning might involve multiple reasoning paths, we go further by designing quantitative metrics. Our analysis reveals that latent reasoning is not merely a superposition of paths, but also serves as a compressed representation of single-step reasoning trajectories.
>
> We hope our response has addressed your concerns and clarified any misunderstandings. We would be happy to continue the discussion and kindly ask that you reconsider the value of our work.
>
> [1] s1: Simple test-time scaling
>
> [2] Qwen3 Technical Report

---

> ### Author Response · Authors · 2025-11-20
>
> Dear Reviewer,
>
> Many thanks again for your thoughtful reviews. We have carefully addressed the main concerns in our rebuttal.
>
> We would like to kindly ask whether our responses have clarified these points and to invite you to reconsider the overall contribution of the paper in light of the new analysis. If anything remains unclear, we are very happy to provide further details and continue the discussion.
>
> Best regards,
>
> The Authors

---

> ### Author Response · Authors · 2025-11-25
>
> Dear Reviewer uoyr,
>
> Thank you sincerely for recognizing the strengths of our paper. We greatly appreciate your thoughtful feedback. Most of your comments highlight areas that require further clarification or elaboration, and we believe these concerns have been thoroughly addressed in our rebuttal.
>
> We are grateful for your time and consideration, and we look forward to any additional feedback you may have. We hope that our responses will assist you in reassessing the paper's contributions and updating your evaluation.
>
> Best regards,
>
> The Authors

---

### Official Review · Reviewer_YNHm · 2025-10-30

**Soundness:** 3
**Presentation:** 2
**Contribution:** 2
**Rating:** 4
**Confidence:** 2

**Summary:**

This paper presents a method for latent reasoning that makes use of the finding that it is easier to make latent vectors which are "soft embeddings" (mixtures of token embeddings), or at least continuous vectors close to these in the latent space. These hidden states can then be further optimized by training to directly generate these tokens with a combination of losses. This results in performance close to (slightly above or below) CoT on easier tasks, and below on harder tasks, but using less tokens.

**Strengths:**

The results are superior to a number of latent reasoning approaches which are the baselines compared to such as COT-SFT (Wei et al., 2022), iCOT (Deng et al., 2024), COCONUT (Hao et al., 2024), CODI (reproduced) (Tan et al., 2025), and CoLaR.
Performance is close to (slightly above or below) CoT on easier tasks (although I think this was similar on related papers, see weaknesses).

**Weaknesses:**

Overall, it seems a very complicated system (and hence also a bit hard to understand), and achieves minor gains over standard CoT, although I appreciate a number of latent reasoning methods are significantly worse than CoT.
While it is admirable to try to make latent reasoning work, I also have a feeling that tying it too much to standard CoT inevitably makes the performance similar to standard CoT, as also shown in the Soft Thinking and Mixture-of-Input papers I think. I note you cite but do not compare to those (although they are also pretty recent)? I think they also showed marginal gains over text CoT, so a similar story, but less complicated...

While the number of latent tokens is less, it’s also worse on harder tasks (Table 2), so I guess you’d have to compare to conventional CoT with less thought tokens, and see how that compares to really understand a bit more the difference.

The performance of COCONUT seems lower than even reported in the original paper, although I suppose this is a different setting? But still, seems a backwards step to put it in a lower performance setting (smaller model, or..?).

grammar:
“To gain a deeper understanding ofing of the latent reasoning process,”
“Latent tokens are designed to be: (1) semantic compactness, ..”

**Questions:**

See weaknesses.

---

> ### Author Response · Authors · 2025-11-15
>
> Thank you very much for your professional assessment. We’re pleased that you recognize our contribution in improving the accuracy of latent reasoning to be on par with that of explicit reasoning. Indeed, this has been a central goal of many researchers in the field over the past six months, beginning with the introduction of COCONUT.
>
> With that in mind, we’d like to take this opportunity to clarify a few potential misunderstandings.
>
> > Overall, it seems a very complicated system (and hence also a bit hard to understand), and achieves minor gains over standard CoT, although I appreciate a number of latent reasoning methods are significantly worse than CoT. While it is admirable to try to make latent reasoning work, I also have a feeling that tying it too much to standard CoT inevitably makes the performance similar to standard CoT, as also shown in the Soft Thinking and Mixture-of-Input papers I think. I note you cite but do not compare to those (although they are also pretty recent)? I think they also showed marginal gains over text CoT, so a similar story, but less complicated...
>
> **The first core question is: why does it appear that latent reasoning offers no significant accuracy improvement over explicit reasoning?**
>
> We believe that, particularly at this stage of research, the evaluation of latent reasoning should not focus on accuracy alone, but should also consider inference efficiency, especially in terms of inference chain length. This is central to the motivation behind COCONUT-style training methods, which aim to significantly reduce the inference chain length while preserving performance that is as close as possible to that of explicit reasoning.
>
> This is precisely what one of our contributions addresses: we make latent reasoning performance comparable to explicit reasoning, while cutting the inference chain length by half.
>
> When should we expect accuracy to significantly exceed that of explicit reasoning? We believe this will become feasible with the advancement of implicit reinforcement learning (RL) methods. Our proposed method, Latent-SFT, is a supervised fine-tuning (SFT)-based approach and does not incorporate RL. As a result, it is intuitive that its performance on high-difficulty reasoning tasks may be relatively limited compared to methods that integrate learning from outcome signals (e.g., through reinforcement learning).
>
> **The second core question is: why are current latent reasoning methods inherently tied to standard chain-of-thought (CoT)?**
>
> As discussed in lines 66–70, latent reasoning lacks direct supervision signals, and most existing approaches distill latent reasoning from explicit CoT reasoning chains. This introduces a fundamental limitation: the upper bound of latent reasoning accuracy becomes that of explicit reasoning, since the model is essentially learning a hidden abstraction of CoT.
>
> However, this limitation maybe can be overcome by incorporating latent reinforcement learning. Since RL does not rely on explicit reasoning traces and allows the model to explore a diverse range of latent paths, it has the potential to break through the performance ceiling imposed by supervised CoT distillation.
>
> **The third core question is: why isn’t our method directly compared to Soft Thinking or Mixture-of-Input?**
>
> This is an excellent question. In fact, around the same time as the Soft-Thinking work, we had already experimented with similar training-free methods, but the results were not very satisfactory. Based on this, we summarize the differences between the two paradigms as follows:
>
> 1. COCONUT-type methods (training-based):
> These approaches aim to shift from the traditional token-based autoregressive paradigm to a hidden-state-based autoregressive paradigm via training. The primary motivation is to reduce inference chain length while preserving model performance. As such, two evaluation metrics are typically considered: accuracy and inference chain length. However, the results are not ideal; most previous methods did not achieve the performance of explicit inference on low-difficulty datasets.
>
> 2. Soft-thinking-type methods (training-free):
> These methods attempt to interpolate between hard and soft tokens without training—usually via weighted embedding techniques. Because they lack a training phase, they cannot reduce the inference chain length and focus solely on the accuracy metric. They are typically evaluated on larger-scale models. The results show that they do not bring significant performance improvements compared to explicit inference.
>
> Our method can be seen as combining the advantages of both paradigms. It is motivated by the goal of reducing inference chain length while maintaining strong model performance. Therefore, our method is fundamentally a COCONUT-style approach (i.e., training-based), and as such,  there is no need to compare it with training-free methods.

---

> ### Author Response · Authors · 2025-11-15
>
> > While the number of latent tokens is less, it’s also worse on harder tasks (Table 2), so I guess you’d have to compare to conventional CoT with less thought tokens, and see how that compares to really understand a bit more the difference.
>
> Thank you very much for the reminder. We sincerely appreciate your constructive suggestion. It’s possible that, due to a less-than-ideal presentation of our results, the accuracy metrics on high-difficulty mathematical reasoning datasets received more attention, while the inference chain length metric—which we believe is equally important—may have been overlooked(which could be one of the reasons behind the paper’s initially low score). In response, we have reorganized our results and added additional Chain-of-Thought (CoT) baselines for a more comprehensive comparison.
>
> It's also important to highlight that our model was trained on sequences of only 4k tokens, which, as those familiar with reasoning tasks will understand, can heavily constrain model performance. To ensure a fair comparison, we evaluated the COT-SFT model with different reasoning-chain lengths and used a fair budget-forcing [1] procedure to obtain the answers. We also follow the evaluation settings NoThink [2] used in Qwen3 to ensure consistency in model assessment. To further unify the evaluation of accuracy and inference efficiency, we report the Pass@1 ⁄ #L (100×) metric, which reflects performance normalized by inference chain length. The experimental results are shown in the table below.
>
> | Model                         | MATH-500 Pass@1              | MATH-500 #L       | MATH-500 Pass@1⁄#(100×)      | AIME24 Pass@1                 | AIME24 #L       | AIME24 Pass@1⁄#(100×)        |
> |------------------------------|------------------------------|-------------------|-------------------|-------------------------------|-----------------|-----------------|
> | COT-SFT                      | 92.6 ± 1.19                  | 3616 ± 232        | 2.56       | 54.4 ± 1.74                   | 12992 ± 346     | 0.42       |
> | COT-SFT (1k-Budget)          | 75.5 ± 1.38               | 1017 ±6.44       | 7.42       | 12.5 ± 1.28                   | 1046 ± 0.01     | 1.19       |
> | COT-SFT (2k-Budget)          | 85.8 ± 1.83               | 1670 ±19.1       | 5.14       | 23.7 ± 1.27                   | 2066 ± 1.08     | 1.15      |
> | COT-SFT (3k-Budget)          | 89.2 ± 0.90               | 2057 ±26.8       | 4.34       | 33.5 ± 1.40                   | 3017 ± 8.06     | 1.11       |
> | COT-SFT (4k-Budget)          | 90.5 ± 0.66               | 2318 ±40.8       | 3.90       | 39.0 ± 1.15                   | 4060 ± 13.7     | 0.96       |
> | COT-SFT (NoThink)          | 77.5 ± 1.88              | 938.5 ±131       | 8.26       | 22.3 ± 1.13                   | 4382 ± 234     | 0.51       |
> | Latent-SFT (Hidden State)    | 67.8 ± 0.94                  | 698.1 ± 22.1      | 9.71       | 7.78 ± 2.08                   | 1509 ± 114      | 0.51       |
> | Latent-SFT (*Soft Embedding*)| 79.8 ± 1.35      | 759.3 ± 36.9      | **10.51**       | 19.2 ± 1.60         | 1484 ± 420      | **1.29**       |
>
> As shown in the table above, **under the same inference chain length, latent reasoning consistently outperforms explicit reasoning on both Math500 and AIME24**. Furthermore, across various baseline inference lengths, our model consistently achieved the highest inference efficiency, as measured by the Pass@1⁄#L (100×) metric. This setup ensures a fair and direct comparison between the two paradigms.
>
> In our experimental design, budget-forcing refers to constraining the model to reason within a specified length limit. To enforce this, we appended the instruction:
> ```text
> "Considering the limited time specified by the user, I will now provide the answer based on the above thinking.\n</think>\n\n"
> ```
> This prompt forces the model to stop reasoning at the desired point. In contrast, NoThink involves simply appending:
> ```text
> "\n</think>\n\n"
> ```
> to the prompt, allowing the model to generate the answer without a predefined stopping point.

---

> ### Author Response · Authors · 2025-11-15
>
> > The performance of COCONUT seems lower than even reported in the original paper, although I suppose this is a different setting? But still, seems a backwards step to put it in a lower performance setting (smaller model, or..?).
>
> This is indeed a detailed and important point. The difference in scores compared to the original COCONUT paper primarily stems from our use of a different base model, which was chosen to ensure fair and consistent comparison across methods.
>
> Specifically, we adopted the experimental setup from the CoLaR paper, which introduced a unified evaluation framework covering methods such as COCONUT and CODI. You may refer to their paper (arXiv:2505.16552v2) and official codebase (GitHub - xiaomi-research/colar).
>
> Therefore, this should not be interpreted as a performance regression. Rather, it reflects a deliberate effort to ensure consistent base models and training datasets across all compared methods. Our code, model checkpoints, and datasets are all available in the anonymous repository.
>
> > grammar: “To gain a deeper understanding ofing of the latent reasoning process,” “Latent tokens are designed to be: (1) semantic compactness, ..”
>
> Thank you for pointing this out. We will correct the grammatical issue in the next version of the PDF.
>
> We would like to reiterate the key contributions of our work:
>
> 1. **From low- to high-difficulty mathematical reasoning:**
> Our model outperforms explicit reasoning on low-difficulty mathematical reasoning datasets and surpasses the previous state-of-the-art latent reasoning method **by over 10%**. We also provide **a comprehensive analysis** of the differences between latent and explicit reasoning on high-difficulty benchmarks, offering valuable insights for the community. Furthermore, we show that our method achieves higher inference efficiency than explicit reasoning in these challenging settings.
>
> 2. **A new definition of latent reasoning:**
> We extend the latent token representation from hidden states to soft embeddings based on preliminary experiments, and we show that soft embeddings consistently outperform hidden states on reasoning datasets of varying difficulty.
>
> 3. **Quantitative analysis of latent reasoning paths:**
> While COCONUT provides a qualitative observation that latent reasoning might involve multiple reasoning paths, we go further by designing quantitative metrics. Our analysis reveals that latent reasoning is not merely a superposition of paths, but also serves as a compressed representation of single-step reasoning trajectories.
>
> Thank you very much for your constructive comments — they are invaluable in helping us improve the quality of our work. We hope our responses have addressed your concerns and clarified any misunderstandings. We kindly ask that you reconsider the overall assessment of our manuscript, and we look forward to further discussion.
>
>
> [1] s1: Simple test-time scaling
>
> [2] Qwen3 Technical Report

---

> ### Author Response · Authors · 2025-11-20
>
> Dear Reviewer,
>
> Many thanks again for your thoughtful reviews. We have carefully addressed the main concerns in our rebuttal.
>
> We would like to kindly ask whether our responses have clarified these points and to invite you to reconsider the overall contribution of the paper in light of the new analysis. If anything remains unclear, we are very happy to provide further details and continue the discussion.
>
> Best regards,
>
> The Authors

---

> ### Author Response · Authors · 2025-11-25
>
> Dear Reviewer YNHm,
>
> Thank you sincerely for recognizing the strengths of our paper. We greatly appreciate your thoughtful feedback. Most of your comments highlight areas that require further clarification or elaboration, and we believe these concerns have been thoroughly addressed in our rebuttal.
>
> We are grateful for your time and consideration, and we look forward to any additional feedback you may have. We hope that our responses will assist you in reassessing the paper's contributions and updating your evaluation.
>
> Best regards,
>
> The Authors

---

### Official Review · Reviewer_Yykd · 2025-11-01

**Soundness:** 3
**Presentation:** 3
**Contribution:** 2
**Rating:** 2
**Confidence:** 3

**Summary:**

This paper presents Latent-SFT, a two-stage learning framework to enable latent reasoning in LLMs. This approach restricts the latent space to the column space of the LLM vocabulary and treats latent reasoning as a superposition over vocabulary probabilities.  The first stage of training involves generating latent tokens with an encoder-decoder architecture using the same LLM. In stage-2, the LLM is directly trained to generate these latent tokens for reasoning.

**Strengths:**

1. The Latent-SFT approach achieves new SoTA numbers on GSM8k and matches the CoT-SFT performance for low-difficulty tasks.

2. Unlike existing approaches, restricting the latent reasoning to the column space of the LLM vocabulary directly addresses the misalignment between latent and token spaces.

3 The paper provides extensive analysis and offers valuable insights into the latent reasoning process.

**Weaknesses:**

1. There is still a significant gap for Latent-SFT compared to CoT-SFT on high difficulty tasks like MATH500 or AIME24.

2. The paper lacks results on other reasoning tasks such as GPQA and LiveCodeBench.

3. The paper mainly demonstrates results on 1B for low-difficulty tasks. Demonstrating results on a larger model can help with showing scalability.

4. How does the proposed approach compare with training free approaches like Soft Thinking in terms of both accuracy and efficiency?

5. The paper lacks training and inference compute efficiency numbers compared to other approaches and the overhead of Stage-1 of learning is not specified.

6. Presentation of results can be improved. Table-1 captions are unclear.

**Questions:**

What is the additional compute cost involved in generating the latent tokens, i.e. stage-1?

---

> ### Author Response · Authors · 2025-11-15
>
> Thank you for your thoughtful feedback on our work. I will now address your concerns.
>
> > There is still a significant gap for Latent-SFT compared to CoT-SFT on high difficulty tasks like MATH500 or AIME24.
>
> It’s possible that, due to a less-than-ideal presentation of our results, the accuracy metrics on high-difficulty mathematical reasoning datasets received more attention, while the inference chain length metric—which we believe is equally important—may have been overlooked. We would like to clarify the following key points:
>
> 1. Latent reasoning is a relatively new research paradigm, distinct from explicit reasoning. Most prior methods have struggled to match explicit reasoning even on low-difficulty benchmarks. Our work first closes that gap by nearly matching or surpassing explicit reasoning on low-difficulty datasets. We then extend the evaluation to more challenging benchmarks—Math500 and AIME24—to explore the comparative strengths and limitations of latent versus explicit reasoning. Notably, previous work rarely evaluated latent reasoning in such high-difficulty settings, whereas our study directly addresses this gap.
>
> 2. As shown in Table 2, although accuracy decreases on high-difficulty datasets, our method significantly reduces the required inference chain length.
>
> 3. On high-difficulty reasoning tasks, our approach outperforms traditional baselines in terms of accuracy. Specifically, we achieve **more than a 10% improvement** in accuracy on both the MATH500 and AIME24 benchmarks.
>
> 4. On low-difficulty datasets, our method achieves performance that is comparable to—or even surpasses—that of explicit reasoning. Compared to the previous state-of-the-art method CoLaR, our approach achieves **an improvement of over 10% in accuracy**.
>
> It's also important to highlight that our model was trained on sequences of only 4k tokens, which, as those familiar with reasoning tasks will understand, can heavily constrain model performance. To ensure a fair comparison, we evaluated the COT-SFT model with different reasoning-chain lengths and used a fair budget-forcing [1] procedure to obtain the answers. We also follow the evaluation settings NoThink [2] used in Qwen3 to ensure consistency in model assessment. To further unify the evaluation of accuracy and inference efficiency, we report the Pass@1 ⁄ #L (100×) metric, which reflects performance normalized by inference chain length. The experimental results are shown in the table below.
>
> | Model                         | MATH-500 Pass@1              | MATH-500 #L       | MATH-500 Pass@1⁄#(100×)      | AIME24 Pass@1                 | AIME24 #L       | AIME24 Pass@1⁄#(100×)        |
> |------------------------------|---------------------|--------------|--------------|-----------------|---------------|-----------------|
> | COT-SFT                      | 92.6 ± 1.19                  | 3616 ± 232        | 2.56       | 54.4 ± 1.74                   | 12992 ± 346     | 0.42       |
> | COT-SFT (1k-Budget)          | 75.5 ± 1.38               | 1017 ±6.44       | 7.42       | 12.5 ± 1.28                   | 1046 ± 0.01     | 1.19       |
> | COT-SFT (2k-Budget)          | 85.8 ± 1.83               | 1670 ±19.1       | 5.14       | 23.7 ± 1.27                   | 2066 ± 1.08     | 1.15      |
> | COT-SFT (3k-Budget)          | 89.2 ± 0.90               | 2057 ±26.8       | 4.34       | 33.5 ± 1.40                   | 3017 ± 8.06     | 1.11       |
> | COT-SFT (4k-Budget)          | 90.5 ± 0.66               | 2318 ±40.8       | 3.90       | 39.0 ± 1.15                   | 4060 ± 13.7     | 0.96       |
> | COT-SFT (NoThink)          | 77.5 ± 1.88              | 938.5 ±131       | 8.26       | 22.3 ± 1.13                   | 4382 ± 234     | 0.51       |
> | Latent-SFT (Hidden State)    | 67.8 ± 0.94                  | 698.1 ± 22.1      | 9.71       | 7.78 ± 2.08                   | 1509 ± 114      | 0.51       |
> | Latent-SFT (*Soft Embedding*)| 79.8 ± 1.35      | 759.3 ± 36.9      | **10.51**       | 19.2 ± 1.60         | 1484 ± 420      | **1.29**       |
>
> As shown in the table above, **under the same inference chain length, latent reasoning consistently outperforms explicit reasoning on both Math500 and AIME24**. Furthermore, across various baseline inference lengths, our model consistently achieved the highest inference efficiency, as measured by the Pass@1⁄#L (100×) metric. This setup ensures a fair and direct comparison between the two paradigms.
>
> In our experimental design, budget-forcing refers to constraining the model to reason within a specified length limit. To enforce this, we appended the instruction:
> ```text
> "Considering the limited time specified by the user, I will now provide the answer based on the above thinking.\n</think>\n\n"
> ```
> This prompt forces the model to stop reasoning at the desired point. In contrast, NoThink involves simply appending:
> ```text
> "\n</think>\n\n"
> ```
> to the prompt, allowing the model to generate the answer without a predefined stopping point.

---

> ### Author Response · Authors · 2025-11-15
>
> > The paper lacks results on other reasoning tasks such as GPQA and LiveCodeBench.
>
> We respectfully believe that this point does not represent a core objective of our paper, nor should it be regarded as a weakness.
>
> 1. As noted earlier, latent reasoning is a newly emerging research direction. While previous methods were primarily evaluated on low-difficulty reasoning datasets, our work extends the evaluation to high-difficulty mathematical reasoning tasks such as Math500 and AIME24. This has already allowed us to gain a more detailed and comprehensive understanding of the differences between latent and explicit reasoning.
>
> 2. Our core motivation lies in proposing new definitions, training methods, and frameworks for parsing latent reasoning paths. As such, it is appropriate to evaluate within existing latent reasoning settings and compare against prior latent reasoning methods. Testing generalization on unrelated benchmarks such as GPQA or LiveCodeBench is not necessary in this context, as these tasks are not closely aligned with our research goals.
>
> We agree these tasks are valuable and may explore them in future work, but they are not essential to validate our current contribution.
>
> > The paper mainly demonstrates results on 1B for low-difficulty tasks. Demonstrating results on a larger model can help with showing scalability.
>
> Yes, we agree with your point; however, larger models tend to achieve near-perfect accuracy on the GSM8k dataset, making it difficult to observe meaningful differences between methods. As noted in our initial response, we reorganized the results of the 7B model on more challenging and widely-used mathematical inference datasets. The results demonstrate that even on high-difficulty benchmarks, our method outperforms explicit inference while maintaining a comparable inference chain length. We believe this provides strong evidence of the scalability of our approach across models of varying sizes.
>
> > How does the proposed approach compare with training free approaches like Soft Thinking in terms of both accuracy and efficiency?
>
> This is an excellent question. In fact, around the same time as the Soft-Thinking work, we had already experimented with similar training-free methods, but the results were not very satisfactory. Based on this, we summarize the differences between the two paradigms as follows:
>
> 1. COCONUT-type methods (training-based):
> These approaches aim to shift from the traditional token-based autoregressive paradigm to a hidden-state-based autoregressive paradigm via training. The primary motivation is to reduce inference chain length while preserving model performance. As such, two evaluation metrics are typically considered: accuracy and inference chain length. However, the results are not ideal; most previous methods did not achieve the performance of explicit inference on low-difficulty datasets.
>
> 2. Soft-thinking-type methods (training-free):
> These methods attempt to interpolate between hard and soft tokens without training—usually via weighted embedding techniques. Because they lack a training phase, they cannot reduce the inference chain length and focus solely on the accuracy metric. They are typically evaluated on larger-scale models. The results show that they do not bring significant performance improvements compared to explicit inference.
>
> Our method can be seen as combining the advantages of both paradigms. It is motivated by the goal of reducing inference chain length while maintaining strong model performance.  Therefore, our method is fundamentally a COCONUT-style approach (i.e., training-based), and as such,  there is no need to compare it with training-free methods.

---

> ### Author Response · Authors · 2025-11-15
>
> > The paper lacks training and inference compute efficiency numbers compared to other approaches and the overhead of Stage-1 of learning is not specified.
>
> Thank you for your reminder. We will include the specific computational costs in the main text.
>
> Since our method does not alter the model architecture, inference-time cost is primarily reflected by the inference chain length, which we already report. Here, we supplement the training time on an 8x A100 GPU setup across both training stages:
>
> | Dataset         | Stage1 (Train Encoder) | Stage1 (Train Decoder) | Stage1 (Joint Training) | Stage 2 |
> |-----------------|------------------------|-------------------------|--------------------------|---------|
> | **GSM8k**       | 3h 30min               | 3h 30min                | 4h 40min                 | 6h 30min |
> | **open-r1 (4k)**| 6h 50min               | 6h 50min                | 8h 30min                 | 12h 10min |
>
>
> These times reflect typical runs and may vary slightly depending on random seeds and system load. We believe this provides a transparent and fair view of the training cost associated with our approach.
>
>
> > Presentation of results can be improved. Table-1 captions are unclear.
> Thank you for your reminder. We will update the caption of Table 1 for clarity and include the previously referenced high-difficulty dataset results as a new table in the main text.
>
> > What is the additional compute cost involved in generating the latent tokens, i.e. stage-1?
> Please refer to the table above for details regarding the training costs.
>
>
> We would like to reiterate the key contributions of our work:
>
> 1. **From low- to high-difficulty mathematical reasoning:**
> Our model outperforms explicit reasoning on low-difficulty mathematical reasoning datasets and surpasses the previous state-of-the-art latent reasoning method **by over 10%**. We also provide **a comprehensive analysis** of the differences between latent and explicit reasoning on high-difficulty benchmarks, offering valuable insights for the community. Furthermore, we show that our method achieves higher inference efficiency than explicit reasoning in these challenging settings.
>
> 2. **A new definition of latent reasoning:**
> We extend the latent token representation from hidden states to soft embeddings based on preliminary experiments, and we show that soft embeddings consistently outperform hidden states on reasoning datasets of varying difficulty.
>
> 3. **Quantitative analysis of latent reasoning paths:**
> While COCONUT provides a qualitative observation that latent reasoning might involve multiple reasoning paths, we go further by designing quantitative metrics. Our analysis reveals that latent reasoning is not merely a superposition of paths, but also serves as a compressed representation of single-step reasoning trajectories.
>
> We hope our response has addressed any confusion or misunderstandings. We look forward to a constructive discussion and kindly request that you reconsider the overall evaluation of our work.
>
> [1] s1: Simple test-time scaling
>
> [2] Qwen3 Technical Report

---

> ### Author Response · Authors · 2025-11-20
>
> Dear Reviewer,
>
> Many thanks again for your thoughtful reviews. We have carefully addressed the main concerns in our rebuttal.
>
> We would like to kindly ask whether our responses have clarified these points and to invite you to reconsider the overall contribution of the paper in light of the new analysis. If anything remains unclear, we are very happy to provide further details and continue the discussion.
>
> Best regards,
>
> The Authors

---

> ### Author Response · Authors · 2025-11-25
>
> Dear Reviewer Yykd,
>
> Thank you sincerely for recognizing the strengths of our paper. We greatly appreciate your thoughtful feedback. Most of your comments highlight areas that require further clarification or elaboration, and we believe these concerns have been thoroughly addressed in our rebuttal.
>
> We are grateful for your time and consideration, and we look forward to any additional feedback you may have. We hope that our responses will assist you in reassessing the paper's contributions and updating your evaluation.
>
> Best regards,
>
> The Authors

---

### Official Review · Reviewer_Q6oS · 2025-11-01

**Soundness:** 2
**Presentation:** 3
**Contribution:** 3
**Rating:** 6
**Confidence:** 3

**Summary:**

The paper addresses the problem that latent reasoning in large language models—reasoning without explicit chain-of-thought tokens—suffers from performance loss due to the unstructured and misaligned nature of latent spaces. To solve this, it proposes Latent-SFT, a two-stage training framework that constrains latent reasoning to the vocabulary embedding space, treating each latent token as a soft superposition of vocabulary vectors. This alignment allows models to reason efficiently and semantically consistently. Experiments show that Latent-SFT achieves explicit-level reasoning accuracy on benchmarks like GSM8K while reducing reasoning chain length by up to 4×, establishing a new state of the art for latent reasoning.

**Strengths:**

* **Originality:** The work defines latent reasoning as a *vocabulary-space superposition*, offering a fundamentally new and interpretable view of how LLMs reason internally.
* **Quality:** The work proposes a well-justified two-stage **Latent-SFT** framework that effectively aligns latent tokens with the LLM’s vocabulary manifold.
* **Significance:** The work demonstrates explicit-level reasoning accuracy with up to **4× shorter reasoning chains**, improving both efficiency and interpretability.

**Weaknesses:**

* **Limited evaluation scope:** Experiments are primarily focused on mathematical reasoning datasets (e.g., GSM8K, Math500, AIME24), leaving uncertainty about generalization to other reasoning domains such as commonsense, symbolic, or multi-modal reasoning. Including ARC may be helpful.
* **High-difficulty performance gap:** Latent-SFT shows substantial accuracy degradation on long-chain datasets like AIME24, suggesting the method struggles to maintain semantic integrity under extended reasoning or limited training lengths; further scaling or curriculum-based training could help.
* **Limited interpretability validation:** Although the paper claims interpretability via “vocabulary-space superposition,” the analysis (ECR@K and N_eff) remains indirect; more qualitative visualization or alignment studies between latent and explicit reasoning paths could make the claim more convincing.
* **Ablation and baseline breadth:** The ablations mostly vary internal components (e.g., LTIM, LTSuM, hidden-state variants) but do not compare against broader compression or distillation methods (e.g., TokenSkip, LoRA-compressed CoT), limiting the contextual strength of performance claims.

**Questions:**

See weakness

---

> ### Author Response · Authors · 2025-11-15
>
> Thank you for your thoughtful feedback on our work. I will now address your concerns point by point.
>
> > 1. Limited evaluation scope: Experiments are primarily focused on mathematical reasoning datasets (e.g., GSM8K, Math500, AIME24), leaving uncertainty about generalization to other reasoning domains such as commonsense, symbolic, or multi-modal reasoning. Including ARC may be helpful.
>
> We believe our evaluation scope in the field of latent reasoning is already quite comprehensive, for the following reasons:
>
> 1. Most existing latent reasoning methods [1-3] focus exclusively on low-difficulty mathematical datasets such as GSM8k. In contrast, we expand the evaluation to include high-difficulty reasoning benchmarks like Math500 and AIME24, offering **broader and more rigorous coverage** than prior work.
>
> 2. Mathematical reasoning tasks serve as a strong proxy for evaluating reasoning capabilities. Since latent reasoning methods currently do not yet match the performance of explicit reasoning on these mathematical tasks, **extending the evaluation to other reasoning domains would be premature** and somewhat misaligned with our core motivation.
>
> In summary, we argue that using both low- and high-difficulty mathematical reasoning datasets provides a sufficient and meaningful basis for comparison between latent and explicit reasoning. Including datasets from other domains, while possible in future work, is **not essential at this stage**.

---

> ### Author Response · Authors · 2025-11-15
>
> > 2. High-difficulty performance gap: Latent-SFT shows substantial accuracy degradation on long-chain datasets like AIME24, suggesting the method struggles to maintain semantic integrity under extended reasoning or limited training lengths; further scaling or curriculum-based training could help.
>
> We would like to emphasize that evaluation should not focus solely on accuracy; **inference chain length** is also a critical factor. Furthermore, while most prior latent reasoning methods have focused exclusively on low-difficulty mathematical reasoning datasets, our comprehensive analysis of the differences between latent and explicit reasoning on high-difficulty benchmarks marks a advancement in the field. We would also like to emphasize the following points for further clarity and context:
>
> 1. As shown in Table 2, although accuracy decreases on high-difficulty datasets, our method significantly reduces the required inference chain length.
>
> 2. On high-difficulty reasoning tasks, our approach outperforms traditional baselines in terms of accuracy. Specifically, we achieve **more than a 10% improvement** in accuracy on both the MATH500 and AIME24 benchmarks.
>
> 3. On low-difficulty datasets, our method achieves performance that is comparable to—or even surpasses—that of explicit reasoning. Compared to the previous state-of-the-art method CoLaR, our approach achieves **an improvement of over 10% in accuracy**.
>
> It's also important to highlight that our model was trained on sequences of only 4k tokens, which, as those familiar with reasoning tasks will understand, can heavily constrain model performance. To ensure a fair comparison, we evaluated the COT-SFT model with different reasoning-chain lengths and used a fair budget-forcing [4] procedure to obtain the answers. We also follow the evaluation settings NoThink [5] used in Qwen3 to ensure consistency in model assessment. To further unify the evaluation of accuracy and inference efficiency, we report the Pass@1 ⁄ #L (100×) metric, which reflects performance normalized by inference chain length. The experimental results are shown in the table below.
>
> | Model                         | MATH-500 Pass@1              | MATH-500 #L       | MATH-500 Pass@1⁄#(100×)      | AIME24 Pass@1                 | AIME24 #L       | AIME24 Pass@1⁄#(100×)        |
> |------------------------------|------------------------------|-------------------|-------------------|-------------------------------|-----------------|-----------------|
> | COT-SFT                      | 92.6 ± 1.19                  | 3616 ± 232        | 2.56       | 54.4 ± 1.74                   | 12992 ± 346     | 0.42       |
> | COT-SFT (1k-Budget)          | 75.5 ± 1.38               | 1017 ±6.44       | 7.42       | 12.5 ± 1.28                   | 1046 ± 0.01     | 1.19       |
> | COT-SFT (2k-Budget)          | 85.8 ± 1.83               | 1670 ±19.1       | 5.14       | 23.7 ± 1.27                   | 2066 ± 1.08     | 1.15      |
> | COT-SFT (3k-Budget)          | 89.2 ± 0.90               | 2057 ±26.8       | 4.34       | 33.5 ± 1.40                   | 3017 ± 8.06     | 1.11       |
> | COT-SFT (4k-Budget)          | 90.5 ± 0.66               | 2318 ±40.8       | 3.90       | 39.0 ± 1.15                   | 4060 ± 13.7     | 0.96       |
> | COT-SFT (NoThink)          | 77.5 ± 1.88              | 938.5 ±131       | 8.26       | 22.3 ± 1.13                   | 4382 ± 234     | 0.51       |
> | Latent-SFT (Hidden State)    | 67.8 ± 0.94                  | 698.1 ± 22.1      | 9.71       | 7.78 ± 2.08                   | 1509 ± 114      | 0.51       |
> | Latent-SFT (*Soft Embedding*)| 79.8 ± 1.35      | 759.3 ± 36.9      | **10.51**       | 19.2 ± 1.60         | 1484 ± 420      | **1.29**       |
>
> As shown in the table above, **under the same inference chain length, latent reasoning consistently outperforms explicit reasoning on both Math500 and AIME24**. Furthermore, across various baseline inference lengths, our model consistently achieved the highest inference efficiency, as measured by the Pass@1⁄#L (100×) metric. This setup ensures a fair and direct comparison between the two paradigms.
>
> In our experimental design, budget-forcing refers to constraining the model to reason within a specified length limit. To enforce this, we appended the instruction:
> ```text
> "Considering the limited time specified by the user, I will now provide the answer based on the above thinking.\n</think>\n\n"
> ```
> This prompt forces the model to stop reasoning at the desired point. In contrast, NoThink involves simply appending:
> ```text
> "\n</think>\n\n"
> ```
> to the prompt, allowing the model to generate the answer without a predefined stopping point.

---

> ### Author Response · Authors · 2025-11-15
>
> > Limited interpretability validation: Although the paper claims interpretability via “vocabulary-space superposition,” the analysis (ECR@K and N_eff) remains indirect; more qualitative visualization or alignment studies between latent and explicit reasoning paths could make the claim more convincing.
>
> We would like to clarify that our paper does not aim to achieve interpretability of latent reasoning. Rather, our primary contribution is to propose a definition of latent reasoning. Therefore, interpretability is not the focus of this work.
>
> The purpose of introducing the ECR@K and N_eff metrics is to provide a quantitative framework for analyzing latent reasoning paths. These metrics are not intended as tools for interpretability, but rather for measuring properties of the reasoning process. For a visual comparison between latent and explicit reasoning paths, please refer to Figure 9 in the appendix.
>
> > Ablation and baseline breadth: The ablations mostly vary internal components (e.g., LTIM, LTSuM, hidden-state variants) but do not compare against broader compression or distillation methods (e.g., TokenSkip, LoRA-compressed CoT), limiting the contextual strength of performance claims.
>
> Our paper primarily focuses on the development of latent reasoning methods. As such, we believe that comparisons with other latent reasoning approaches and with explicit reasoning baselines are sufficient to support our claims. Specifically, we propose a new framework for defining, training, and resolving latent reasoning. Demonstrating improvements over prior latent reasoning methods is logically consistent with the goals of our paper and does not require direct comparisons to non-latent reasoning methods such as TokenSkip or LoRA-compressed CoT.
>
> That said, we acknowledge the relevance of these related approaches, and we plan to conduct a more comprehensive investigation—including comparisons with non-latent methods—in future work.
>
> We would like to reiterate the key contributions of our work:
>
> 1. **From low- to high-difficulty mathematical reasoning:**
> Our model outperforms explicit reasoning on low-difficulty mathematical reasoning datasets and surpasses the previous state-of-the-art latent reasoning method **by over 10%**. We also provide **a comprehensive analysis** of the differences between latent and explicit reasoning on high-difficulty benchmarks, offering valuable insights for the community. Furthermore, we show that our method achieves higher inference efficiency than explicit reasoning in these challenging settings.
>
> 2. **A new definition of latent reasoning:**
> We extend the latent token representation from hidden states to soft embeddings based on preliminary experiments, and we show that soft embeddings consistently outperform hidden states on reasoning datasets of varying difficulty.
>
> 3. **Quantitative analysis of latent reasoning paths:**
> While COCONUT provides a qualitative observation that latent reasoning might involve multiple reasoning paths, we go further by designing quantitative metrics. Our analysis reveals that latent reasoning is not merely a superposition of paths, but also serves as a compressed representation of single-step reasoning trajectories.
>
> We hope our responses have adequately addressed your concerns. We look forward to a constructive dialogue and respectfully invite you to reassess the quality of our work in light of these clarifications.
>
>
> [1] Training Large Language Models to Reason in a Continuous Latent Space
>
> [2] Think Silently, Think Fast: Dynamic Latent Compression of LLM Reasoning Chains
>
> [3] CODI: Compressing Chain-of-Thought into Continuous Space via Self-Distillation
>
> [4] s1: Simple test-time scaling
>
> [5] Qwen3 Technical Report

---

> ### Author Response · Authors · 2025-11-20
>
> Dear Reviewer,
>
> Many thanks again for your thoughtful reviews. We have carefully addressed the main concerns in our rebuttal.
>
> We would like to kindly ask whether our responses have clarified these points and to invite you to reconsider the overall contribution of the paper in light of the new analysis. If anything remains unclear, we are very happy to provide further details and continue the discussion.
>
> Best regards,
>
> The Authors

---

> ### Author Response · Authors · 2025-11-25
>
> Dear Reviewer Q6oS,
>
> Thank you sincerely for recognizing the strengths of our paper. We greatly appreciate your thoughtful feedback. Most of your comments highlight areas that require further clarification or elaboration, and we believe these concerns have been thoroughly addressed in our rebuttal.
>
> We are grateful for your time and consideration, and we look forward to any additional feedback you may have. We hope that our responses will assist you in reassessing the paper's contributions and updating your evaluation.
>
> Best regards,
>
> The Authors

---

### Author Response · Authors · 2025-12-01
**Discussion Summary and Clarifications**

Dear AC,

Thank you very much for your time to ICLR 2026. We sincerely hope you might consider revisiting our submission. We believe that the initial low score may be due to **a misunderstanding of the paper’s core contribution** — a concern supported by the reviewers’ **average confidence score of only 2.75**.

Specifically, most reviewers focused solely on the drop in Pass@1 accuracy on AIME24 and Math500, while **overlooking inference efficiency metric**.  We would also like to clarify the accuracy issue:

> Due to limited computational resources, our model was trained with **4k context length**, whereas explicit SFT typically uses **32k**. When evaluated under the same 4k budget, our method outperforms explicit SFT on Pass@1, demonstrating its practical strength under fair conditions.

In addition, they also raised some experimental requirements that were not very relevant to the motivation of our paper. We believe these are **not necessary and should not be considered weaknesses**, therefore we rejected them. We respectfully disagreed with those points in our rebuttal, submitted on **November 15**, and we believe we have thoroughly addressed all substantive concerns. However, **as of today**, we have not received any response from the reviewers.



We would like to reiterate the key contributions of our work:

**From low- to high-difficulty mathematical reasoning:** While most existing work on latent reasoning focuses on performance on low-difficulty mathematical reasoning datasets, our study extends the evaluation to high-difficulty benchmarks. Our model outperforms explicit reasoning on low-difficulty mathematical reasoning datasets and surpasses the previous state-of-the-art latent reasoning method by over 10%. We also provide a comprehensive analysis of the differences between latent and explicit reasoning on high-difficulty benchmarks, offering valuable insights for the community. Furthermore, we show that our method achieves higher inference efficiency than explicit reasoning in these challenging settings.

**A new definition of latent reasoning:** We extend the latent token representation from hidden states to soft embeddings based on preliminary experiments, and we show that soft embeddings consistently outperform hidden states on reasoning datasets of varying difficulty.

**Quantitative analysis of latent reasoning paths:** While COCONUT provides a qualitative observation that latent reasoning might involve multiple reasoning paths, we go further by designing quantitative metrics. Our analysis reveals that latent reasoning is not merely a superposition of paths, but also serves as a compressed representation of single-step reasoning trajectories.

We believe our work advances the state of the art in latent reasoning, and that the proposed Latent-SFT framework establishes a solid foundation for future developments in latent reinforcement learning (Latent RL).

Thank you again for your time and support.

Warm regards,

Authors.

---

### Meta-Review · Area_Chair_kZbw · 2026-01-08

**Summary:**

1. The method seems to be quite complicated and needs ablations. Reviewers complain that the steps are not sufficiently transparant.
2. Need to compare with similar approaches (e.g., soft-thinking / mixture style methods).
3. Still a big gap between the proposed approach and CoT, for the difficult tasks (e.g., AIME).

**Reviewer Concerns:**

Some of the issues are addressed, but many remain. The paper needs to be improved on both clarity and experimental results. It would be great if multiple stages of the proposed methods can be simplified (or merged).

**Reviewer Scores:**

Given the rebuttal, the reviewers likely keep their scores.

---

### Decision · Program_Chairs · 2026-01-26

Reject